# Biofunctionalized dissolvable hydrogel microbeads enable efficient characterization of native protein complexes

Xinyang Shao [1,2,10], Meng Tian [3,4,10], Junlong Yin[3], Haifeng Duan[5], Ye Tian[2], Hui Wang[6], Changsheng Xia[6], Ziwei Wang[3], Yanxi Zhu[7], Yifan Wang[7,8], Lingxiao Chaihu[1,9], Minjie Tan[1], Hongwei Wang [3,4], Yanyi Huang [1,2,7,8], Jianbin Wang [2,3] ✉ & Guanbo Wang [1,7] ✉

The characterization of protein complex is vital for unraveling biological mechanisms in various life processes. Despite advancements in biophysical tools, the capture of non-covalent complexes and deciphering of their bio-chemical composition continue to present challenges for low-input samples. Here we introduce SNAP-MS, a Stationary-phase-dissolvable Native Affinity Purification and Mass Spectrometric characterization strategy. It allows for highly efficient purification and characterization from inputs at the pico-mole level. SNAP-MS replaces traditional elution with matrix dissolving during the recovery of captured targets, enabling the use of high-affinity bait-target pairs and eliminates interstitial voids. The purified intact protein complexes are compatible with native MS, which provides structural information including stoichiometry, topology, and distribution of proteoforms, size variants and interaction states. An algorithm utilizes the bait as a charge remover and mass corrector significantly enhances the accuracy of analyzing heterogeneously glycosylated complexes. With a sample-to-data time as brief as 2 hours, SNAP-MS demonstrates considerable versatility in characterizing native complexes from biological samples, including blood samples.

Protein complexes are central to executing biological functions across various life processes. The analysis of these intact protein complexes is pivotal to understanding biological processes at the molecular level. Over the past decades, methodological advancements have drawn significant attention to the characterization of protein complexes and the deciphering of biological mechanisms. This includes the analysis of the subunit identities, binding stoichiometry, mutations, diverse post-translational modifications (PTMs), complex topology, and dynamic higher-order structures[1]. While such characterizations can be conducted using biophysical techniques such as nuclear magnetic resonance (NMR) spectroscopy[2–4], X-ray crystallography[5] and electron microscopy (EM)[6], these methods often require supplementary techniques to overcome their limitations, which may include demands on the

[1]Institute of Chemical Biology, Shenzhen Bay Laboratory, Shenzhen, China. [2]Changping Laboratory, Beijing, China. [3]School of Life Sciences, Tsinghua University, Beijing, China. [4]State Key Laboratory of Membrane Biology, Beijing Frontier Research Center of Biological Structures, Tsinghua-Peking Joint Center for Life Sciences, Tsinghua University, Beijing, China. [5]CYGNUS Bioscience (Beijing) Co. Ltd, Beijing, China. [6]Department of Clinical Laboratories, Peking University People's Hospital, Beijing, China. [7]Biomedical Pioneering Innovation Center (BIOPIC), Peking University, Beijing, China. [8]College of Chemistry and Molecular Engineering, Beijing National Laboratory for Molecular Sciences, Peking University, Beijing, China. [9]School of Chemistry & Materials Science, Nanjing Normal University, Nanjing, Jiangsu, China. [10]These authors contributed equally: Xinyang Shao, Meng Tian. ✉e-mail: jianbinwang@tsinghua.edu.cn; guanbo.wang@pku.edu.cn

purity, quantity or crystallizability of the sample. Mass spectrometry (MS) offers a solution to these challenges by allowing the detection, identification, quantitation and structural analysis of proteins or their fragments from solution phase in the presence of complex background molecules. Unlike conventional MS approaches that rely on measurements under denaturing conditions, native MS enables the direct measurement of intact protein complexes under native-like conditions, preserving the non-covalent interactions of proteins. When native MS is combined with tandem MS using appropriate gas-phase dissociation reactions, it can cleave either non-covalent interactions or covalent peptide bonds. This releases subunits (in the "complex-down" approach) or peptide fragments (in the "native top-down" approach) from the complexes, revealing the complex's stoichiometry, topology, subunit's identification or functional sites[7–15]. Native MS has demonstrated its capacity to bridge the gaps of conventional biophysical techniques in fields such as structural biology[4] and drug development[16]. It provides unique information on the structures and dynamics of protein complexes, including protein-lipid[17], protein-nucleic acid[18–20] and protein-protein assemblies[21–23], as well as macromolecular machineries[24,25].

Despite the capability of MS measurements to tolerate a certain quantity of background species, targeted analysis of specific native proteins or complexes from biological samples, such as serum and cell lysates, necessitates the purification of these species to reduce the signal interference and required dynamic range. While various separation methods based on differing physiochemical properties, such as size or charge, do not offer satisfactory separation of such complex systems, conventional native protein purification methods based on affinity purification (AP) allow for capture of target proteins with high specificity. However, these methods often encounter challenges arising from the dilemma between the binding affinity and elution efficiency[26]. Specifically, a higher affinity between the bait-target pair hampers competitive elution under non-denaturing conditions, inevitably leading to a lower recovery efficiency. As a result, the recovery of targets at modest or low abundances often requires a high-input sample and laborious, time-consuming elution using a large among of eluent. Although competitive elution can be substituted with linker cleavage on either the target protein side (*e.g.*, IMPACT intein tag) or purification matrix side (*e.g.*, EZ-LINK cleavable linker), the performance of this approach still requires further improvement due to the unsatisfactory density of bait loading and accessibility of target/cleaver molecules to bead's surface. The substitution of a solid matrix with hydrogel materials (*e.g.*, PC alkyne agarose) improves reagent accessibility and cleavage efficiency. However, the non-specific adsorption of background proteins at the beads' surface, as well as the interstitial void (spaces between the spherical beads), still necessitates elution and limits the efficiency of recovering low-input samples. Therefore, the development of a highly efficient purification method on a small reaction scale remains a challenge.

Another impediment to the direct structural characterization of natural proteins is the high heterogeneity caused by PTMs, particularly glycosylation. As the most abundant and complex type of PTM, glycosylation contributes to a variety of biological processes and is prevalent in natural proteins and their complexes in biological samples. As the carbohydrate content increases, heterogeneously glycosylated proteins exhibit broad mass distributions, leading to highly convoluted signals due to the overlap of broadened signal clusters from different charge states[27]. Furthermore, different proteoforms display non-identical charge state distributions due to their varying charging responses to ionization. This leads to inconsistencies among the mass profiles in individual charge states, invalidating the basic assumption used in conventional deconvolution algorithms[28]. These effects compromise the accuracy and even feasibility of mass measurement and identification of the detected glycosylated species. Previously developed approaches to measure heterogeneous glycoproteins, such as

charge-detection MS (CDMS)[29,30], limited charge reduction[28], and "heterogeneity dilution"[27], rely on either limited models of instruments with functions allowing for the detection of single particle signal or electron/proton-based charge manipulation, or the availability of a cognate monoclonal antibody (Ab). These factors greatly limit the applicability of these approaches.

In this work, we developed a strategy for Stationary-phase-dissolvable Native Affinity Purification and Mass Spectrometric characterization (SNAP-MS) of target protein complexes, aiming to provide an effective solution to these challenges. This strategy is based on development of biofunctionalized dissolvable hydrogel microbeads (SNAP beads), and native MS methods for the purified target systems. Upon capturing of target proteins with high-affinity baits conjugated with the SNAP beads, the process of dissolving the bead matrix releases target in the form of bait-target complex and replaces the need for competitive elution or linker cleavage from conventional beads, thereby enabling the use of high-affinity bait-target pairs while ensuring efficient recovery of target proteins. The volume scale of the purification system is minimized by eliminating the eluent, the interstitial void, and the absorbent surface of the beads. These features facilitate the purification of a variety of proteins with a starting input at the pico-mole level. The output is compatible with downstream native MS for characterizing their binding stoichiometry, topology, and proteoform-specific PTMs. The bait attached to the purified targets can be released through collisional dissociation in the gas phase after being subjected to MS analysis. By leveraging the defined masses of bait molecules and the asymmetric charge partitioning effect upon bait releasing, the accuracy of analyzing heterogeneous glycoproteins and their complexes can be significantly improved. This is achieved through a tandem-MS-based "fragment complementation" approach that utilizes the mass balance constraint and benefits of charge reduction[31]. As a versatile strategy, SNAP-MS can be implemented either independently or in conjunction with conventional tools such as cryo-EM for a more detailed structural characterization of a variety of native protein complexes.

## Results
### Improving purification efficiency through dissolving of bead matrix

To address the challenges associated with competitive elution under native conditions, we developed the SNAP-MS workflow, which incorporates an elution-free affinity purification facilitated by through a bead dissolving strategy (Fig. 1a). Upon capturing target proteins or protein complexes from biological samples via specific bait-target binding with SNAP beads, the traditional competitive elution is supplanted by bead matrix dissolving to recover the targets. The SNAP beads are hydrogel-based microspheres. Both the side-chains connecting the beads with baits (Linker **1**) and the crosslinkers for the linear hydrogel polymer chains (Linker **2**) are designed to be cleavable either through chemical treatment or UV irradiation. Following bead dissolving, the targets are released in the form of bait-target complexes in solution, ready for subsequent characterization using native (tandem) MS or other structural biology techniques such as cryo-EM. This approach, which circumvents competitive elution, allows for a more flexible selection of bait-target pairs for target capture, including those with high affinities that preclude recovery of captured targets through competitive elution. The fabrication process of SNAP beads is elucidated in Fig. 1b. For the chemically dissolvable format of SNAP beads, we synthesized a chemically cleavable Linker **1** (Supplementary Fig. 1a; refer to Supplementary Fig. 2 for $^1$H-NMR and MS characterization) and utilized N, N'-Bis(acryloyl) cystamine as the cleavable Linker **2**. Both linkers contain disulfide bonds that facilitate bead matrix dissolving and release of bait-target complexes under reducing conditions. For the photo-dissolvable format of SNAP beads, we

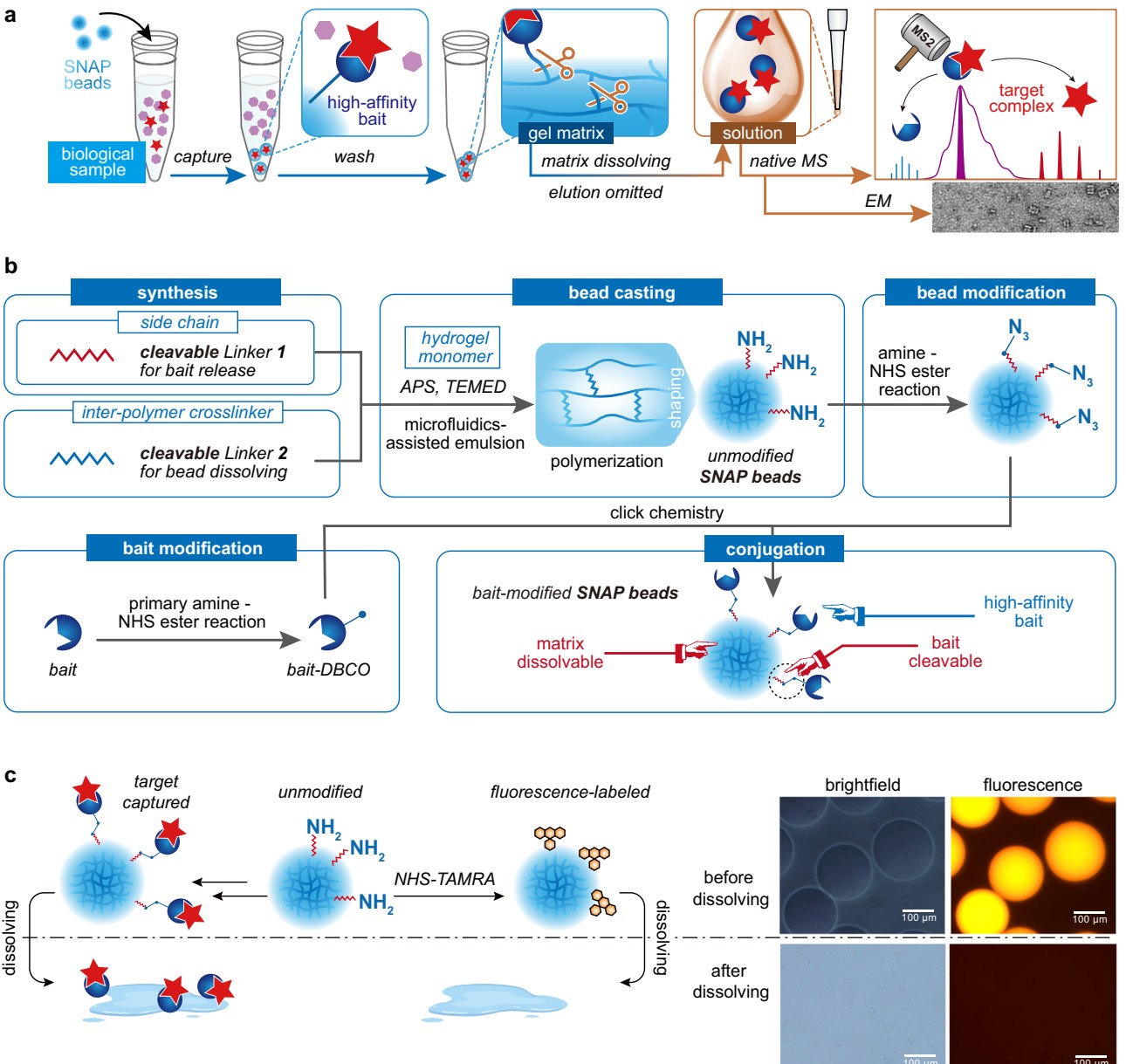

**Fig. 1 | Schematic overview of SNAP-MS. a** Protein purification and characterization workflow utilizing SNAP beads. **b** Fabrication procedure for SNAP beads. **c** Visualization of the dissolving process of fluorophore-labeled SNAP beads, characterized by both brightfield and fluorescence microscopy.

synthesized an o-nitrophenyl-based Linker **2** (Supplementary Fig. 1b; refer to Supplementary Figs. 3 and 4 for ¹H-NMR and MS characterization of key intermediate and product) and employed the light-sensitive PC-Alkyne-PEG4-NHS ester as Linker **1**. This allows bead dissolving through irreversible cleavage of this light-sensitive crosslinker under UV irradiation, thereby avoiding the use of disulfide reductant for protein systems that are incompatible with reducing conditions. Through the generation of precursor mixture droplets via microfluidics-assisted emulsion[32] and subsequent polymerization triggered with ammonium persulfate and tetramethylethylenediamine (APS-TEMED), the hydrogel was formed into microbeads of defined sizes (Supplementary Fig. 5). After modifying the bait protein with a DBCO group (Supplementary Fig. 6) and the bead with an azide group (Supplementary Fig. 7), the beads are conjugated with the bait proteins through click-chemistry (Supplementary Fig. 7). Depending on the chemical nature of the exposed functional groups, the bead surface may exhibit different zeta potentials upon modification or bait-conjugation (Supplementary Fig. 8). The performance of SNAP beads

upon chemical or photo-treatment was evaluated using brightfield and fluorescent microscopy of SNAP beads modified with the fluorophore 5-(and 6-)carboxytetramethylrhodamine, succinimidyl ester (TAMRA-NHS). This revealed the elimination of the spherical beads (Fig. 1c and Supplementary Fig. 9 for chemically dissolvable and photo-dissolvable SNAP beads, respectively). The shift in bead size distribution in response to dissolving treatment, as revealed by dynamic light scattering (DLS), confirmed the complete dissolving of SNAP beads (Supplementary Fig. 10).

The benefits of the bead dissolving strategy can be illustrated by comparing the performance of SNAP beads with other existing bead types in purifying a model protein, green fluorescent protein (GFP) (Fig. 2). We utilized SNAP beads conjugated with streptavidin (SA), commercial agarose beads conjugated with Strep-Tactin (ST), and magnetic beads (M280) conjugated with SA to capture biotin-modified GFP (Supplementary Fig. 6c). Both SA and ST are commonly used as affinity baits for biotin-tagged targets. Following the capture of GFP, the SNAP beads were treated with TCEP to release the captured GFP via

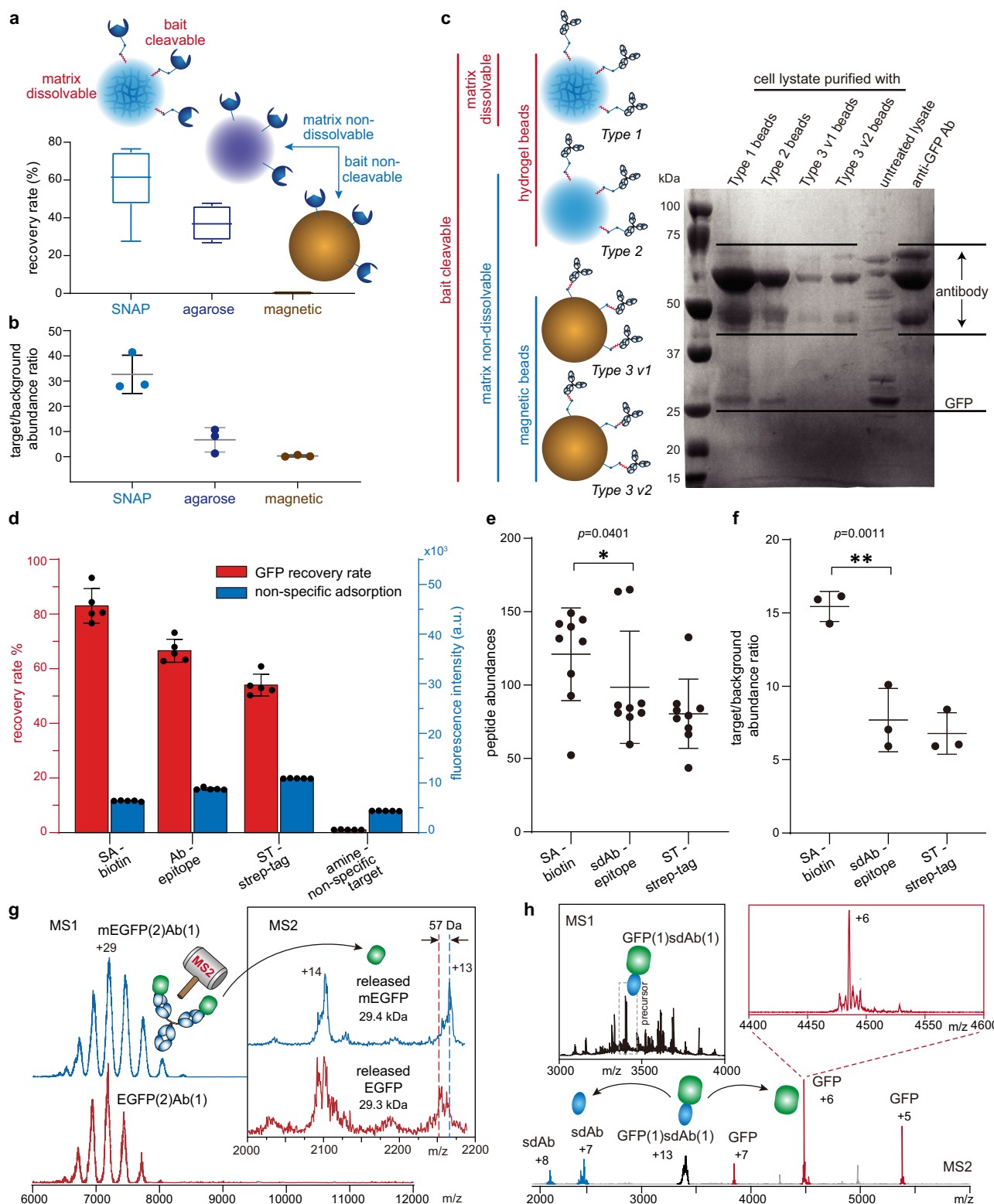

bead dissolving. In contrast, the agarose and magnetic beads were treated with a commercial elution buffer (Buffer E) containing desthiobiotin to recover GFP. The recovery rate was determined based on the fluorescence of GFP, while the purification specificity was evaluated by measuring the intensity ratio of the target GFP to background proteins using MS-based label-free quantitative proteomics (Fig. 2a). The magnetic beads were unable to recover the target due to the high-affinity binding between SA and biotin, which prevented the release of the biotin-modified GFP through competitive elution under non-

denaturing conditions. The agarose beads conjugated with ST, a relatively weaker biotin-binder, achieved modest target recovery through competitive elution. However, this came at the expense of low specificity due to substantial non-specific adsorption of background proteins at the bead surface. In comparison to these non-dissolvable and bait-non-cleavable beads, the SNAP beads demonstrated superior performance, exhibiting not only higher recovery but also higher specificity (Fig. 2a; the background proteins were modified with fluorescent label TAMRA-NHS for fluorescence detection, followed by

**Fig. 2 | Performance evaluation of SNAP-MS in purifying GFP from *E. coli* lysate.** **a** Recovery rate of biotin-modified GFP purified using SNAP beads conjugated with SA, agarose beads conjugated with ST (ST XT), and magnetic beads (M280) conjugated with SA. The box-and-whisker plots illustrate extreme values using the whiskers, the median at the center of the box, and the 75th and 25th percentiles at the box bounds ($n = 6$). **b** Specificity of purifying biotin-modified GFP with the aforementioned beads. Datapoints represent the average abundance ratios of GFP peptides to background protein peptides ($n = 3$). **c** SDS-PAGE of GFP purified using different types of bait-cleavable beads conjugated with anti-GFP Ab. Type 1: SNAP beads; Type 2: non-dissolvable bait-cleavable hydrogel beads; Type 3: bait-cleavable magnetic beads (two versions differ by conjugation reactions and cleavable site-bead distances). Consistent results were yielded by three independent measurements. **d** Recovery rate and non-specific adsorption of SNAP beads conjugated with SA, anti-GFP Ab and ST in purification of Strep-tagged GFP (10 ng/μL in cell lysate) modified with biotin ($n = 5$). **e** Abundances of peptides from biotin-modified GFP samples purified using SNAP beads conjugated with SA, sdAb and ST respectively. Datapoints represent the detected abundances of different peptides (*: one-way ANOVA test, $p \leq 0.05$). **f** Specificity of purifying biotin-modified GFP with the aforementioned SNAP beads. Datapoints represent the average abundance ratios of GFP peptides to background protein peptides (**: one-way ANOVA test, $p \leq 0.01$). **g** Native intact mass (MS1) spectra of purified EGFP (red) and mEGFP (blue) in complex with the Ab bait. The inset shows the tandem mass (MS2) spectra of EGFP and mEGFP released from the bait-target complexes. **h** MS1 spectrum (top left) of purified GFP in complex with the sdAb bait (polyclonal), and MS2 spectrum (bottom) showing release of GFP from the mass-selected GFP-sdAb complex (top right). Results in (**a**) and (**d**) were determined with fluorescence spectroscopy. Results in (**b**), (**e**) and (**f**) were determined with label-free quantitative proteomics. Data are presented as mean ± SD of independent replicates in this figure. Source data are provided as a Source Data file.

removal of the excess fluorescent molecules through centrifugal ultrafiltration).

Bait-cleavable non-dissolvable beads, a distinct class of existing beads designed for the purification of low-input samples, enable target recovery through the release of bait-target complexes from the bead matrix, eliminating the need for competitive elution. To assess whether bead dissolving provides additional advantages to this bait-cleaving scheme, we compared SNAP beads (Type 1 beads; Supplementary Fig. 7a, Step b2) with their non-dissolvable counterparts, *i.e.*, hydrogel beads fabricated with the same precursor recipe except that the cleavable inter-polymer crosslinker was replaced with a non-cleavable one (Type 2 beads), and magnetic beads modified with cleavable linkers (Type 3 beads). Considering the potential impact of reactive site accessibility on purification performance, we conjugated Type 3 beads with baits through different linkers and conjugation reactions, resulting in different distances between the cleavage sites and the beads (Type 3 v1 and v2 beads; Supplementary Fig. 7d). To emulate the real application scenario of purifying natural proteins without any purification tag from biological samples, we employed an anti-GFP antibody (Ab) as the bait for all these bead types to capture unmodified GFP from cell lysate (Fig. 2b). As revealed by the SDS-PAGE results, higher abundances of Ab were detected in products purified by SNAP beads and Type 2 beads than those by Type 3 beads, suggesting a higher efficiency of bait conjugation with the hydrogel beads. A higher abundance of GFP was detected in the product purified by SNAP than that by Type 2 beads, while GFP was hardly detected in products purified by Type 3 beads. The low purification efficiency of the bait-cleavable non-dissolvable beads can be attributed to the relatively low reagent accessibility on their surface, which resulted in less efficient bait conjugation, protein capturing, and linker cleavage. In contrast, SNAP beads incorporate sufficient reactive side chains for bait conjugation during the casting step, significantly enhancing the capacity of bait loading. Its hydrogel nature reduces the impact of the solid-solution interface on the reactions at the bead surface. The dissolving of the bead matrix facilitates exposure of the cleavable sites at the side chains, thereby accelerating the release of the bait-target complexes. Additionally, bead dissolving completely eliminates the interstitial void of beads, which increases the volume of solution required for high recovery of the target and the dilution fold of the output solution in purification with conventional beads.

Bypassing the elution process allows for a more flexible selection of bait from a wider array of candidates, tailored to the specific purification objective or the properties of the target system. This feature facilitates the use of bait-target pairs with high affinities, thereby enhancing purification efficiency. To illustrate the advantages of employing such pairs, we evaluated the purification efficiencies and specificities for a GFP sample, which was Strep-tagged and modified with biotin-NHS. This was conducted using SNAP beads conjugated with SA, ST, anti-GFP Ab, and unconjugated beads with exposed amino groups, respectively, from TAMRA-labeled *E. coli* lysate. As indicated by the fluorescent signals of GFP and those of the fluorophore-labeled background proteins in cell lysate, the GFP recovery rate increased in line with the increasing binding affinities between bait-target pairs, *i.e.* ST-strep tag, Ab-GFP, and SA-biotin. Concurrently, non-specific adsorption of lysate background proteins decreased in this order (Fig. 2c). Single-domain antibody (sdAb), an upgrade of intact Ab, has emerged as a popular bait for affinity purification due to its advantages, including small size, high stability, easier production, as well as high affinity and specificity[33]. In a comparison involving SNAP beads conjugated with sdAb, the high-affinity bait SA outperformed both sdAb and ST in terms of the absolute abundance of the recovered target (Fig. 2d and Supplementary Fig. 11) and specificity (Fig. 2e, f). This underscores the critical role of bait-target affinity in purification efficiency and the advantage of employing high-affinity bait facilitated by bead dissolving.

## Compatibility of recovered protein sample with native MS characterization

Upon dissolving of the SNAP beads, the captured targets are released into the solution in the form of bait-target complex, making them available for the downstream structural characterization. The dissolving product of the bead matrix, the linear polyacrylamide chains, are also present in the solution. While these polymer chains are barely visible and do not interfere with the characterization of protein particles in electron microscopy (EM) (Supplementary Fig. 12), they do generate signals in mass spectrometry (MS) that may interfere with the protein signals (Supplementary Fig. 13). These polymers can be removed through various methods such as centrifugal ultrafiltration, chromatography (*e.g.*, size exclusion (SEC) or ion exchange (IEX)), or in-source collisional fragmentation in the gas phase (Supplementary Fig. 14; refer to Supplementary Table 1 for removal methods for samples tested in this work). In an example of purifying two enhanced GFP variants (wildtype EGFP and its monomeric A206K mutant, mEGFP) overexpressed in *E. coli* from cell lysate using SNAP beads conjugated with anti-GFP antibody (Ab), the purified EGFP variants in complex with the Ab bait were characterized by native MS following bead dissolving and polymer removal through SEC (Fig. 2g). The intact mass (MS1) measurement revealed the binding stoichiometry of the target-bait (EGFP-Ab) complexes. The tandem mass (MS2) measurement facilitated the removal of the bait from the bait-target complexes through collisional dissociation (either collision-induced dissociation, CID, or higher-energy collisional dissociation, HCD). Upon removal of the Ab bait, the two EGFP variants could be distinguished by the measured mass shift of 57 Da, which is in agreement with their theoretical mass difference (inset of Fig. 2g).

Protein complexes tend to preferentially eject smaller and peripheral subunits during collisional dissociation[34], with the ejected subunit often carrying a disproportionately higher charge (relative to

its mass) than the remaining complementary portions[35–38]. The bait-target interaction is typically located peripherally in the bait-target complex. In the instance of EGFP-Ab complex, given that the size of the bait is larger than that of the target (148 kDa vs. 29 kDa), and each bait molecule can capture more than one target to form an EGFP(2)Ab(1) complex, the target EGFP is released as the ejected subunit with a higher charge density. This leaves its complementary subcomplex EGFP(1)Ab(1) as the low-charge-density dissociation product (Supplementary Fig. 15). When a lower charge density is required for the target, or when the target is a non-covalently assembled protein complex whose integrity should be maintained during bait removal, the ejected subunit can be switched to the bait by reducing the size of the bait. In another example of purifying GFP, when we used sdAb as the bait, which is smaller than GFP (16 kDa vs. 27 kDa), the sdAb bait was ejected as the high-charge-density subunit from the bait-target complex. This left the target GFP in reduced charge states (Fig. 2h).

## Preservation of quaternary structures and proteoform distribution of targets

Glycosylation is prevalent in proteins and their complexes in biological samples. To assess the efficacy of SNAP-MS in preserving the quaternary structures and proteoform distribution of targets, we chose two forms of avidin as models: natural egg white avidin (*eg* avidin) and recombinant avidin expressed from corn (*r* avidin). These are tetrameric glycoprotein complexes, and the glycosylation pattern of the monomer can be adequately resolved in high-resolution mass spectra. Despite having identical amino acid sequences, *r* avidin and *eg* avidin exhibit different PTM patterns and thus different mass distributions. We employed SNAP beads conjugated with biotin to purify these avidins, which were pre-spiked in *E. coli* lysate. Upon bead dissolving, the target avidins were released into the solution as complexes with the biotin bait (Fig. 3a). In the native MS characterization of the bait-target complexes (Fig. 3b), the recovered avidin complexes retained their native tetrameric states without losing any subunit under low collisional dissociation settings. The detected octamers, in addition to tetramers, were shown to be stable species in solution, rather than artifacts introduced during the purification or MS stage, as demonstrated by SEC separation and online SEC-MS measurements (Supplementary Fig. 16). Due to the incorporation of the biotin bait, both the tetramer and octamer of the recovered avidin exhibited a mass increase compared to their original counterparts. Further characterization of the avidin monomer could be achieved through an increase in collisional energy, which resulted in the release of the avidin monomer in addition to the biotin bait (Fig. 3c). The aforementioned mass increase of the recovered oligomers remained unchanged at higher desolvation voltages (Supplementary Fig. 17), suggesting that the origin of such a mass increase did not include inadequate desolvation during ionization.

The high-resolving power of the instrument used in this study enabled the signals of the released avidin monomer ions to be isotopically resolved. This ensured a reliable deconvolution of the spectra, revealing their proteoform distribution profiles. Both the recovered *r* avidin and *eg* avidin aligned with the profiles of their original counterparts (Fig. 3d). This suggests a complete recovery of the proteoform ensemble without introducing bias against any subpopulation during purification. The monomers released through SF could be mass-selected for subsequent tandem MS with collisional dissociation, thereby enabling top-down analysis in a pseudo-MS3 manner. A comparison of fragment ion signals of the recovered and original avidins suggests that the fragmentation pattern was not affected by the purification process (Fig. 3e). Few fragments were detected within the segment between Cys4 and Cys83 of either the original or recovered avidin. This suggests that their Cys4-Cys83 disulfide bond remained intact at the MS stage, even though the captured avidin underwent TCEP treatment during bead dissolution. Indeed, previous studies have shown that chemical disulfide reductants under

non-denaturing conditions have only a minimal effect on protein integrity. Intrachain disulfide bonds are less susceptible to such reductants under native conditions (their reduction often requires denaturing conditions)[39]. Furthermore, even if the interchain disulfide bonds are reduced, quaternary structures can still be preserved by non-covalent interactions[40] (Supplementary Fig. 18a).

## Revealing individual-specific distribution of sizes and binding states of endogenous glycoprotein complexes from human sera

SNAP-MS is well-suited for the efficient purification and characterization of endogenous protein complexes, including heterogeneously glycosylated ones, from blood samples. Haptoglobin (Hp), a plasma protein that binds to hemoglobin (Hb) released from erythrocytes, is a biomarker of interest when considering hemolysis[41]. Hp exhibits significant heterogeneity in terms of polypeptide chain lengths, glycosylation, oligomeric states, and binding states. Current diagnostic methods only detect Hp abundance, but not its oligomeric states and stoichiometry with Hb in complexes, which are more clinically relevant. MS analysis of Hp at the intact level faces challenges due to not only heterogeneity in oligomeric states but also heterogeneity in protomer mass caused by its approximately 20% (m/m) carbohydrate content[31,42]. In this study, we collected blood samples from four donors (P1-P4) who had not been previously diagnosed with any health conditions relevant to this study. We purified Hp from human sera using SNAP beads conjugated with Hb bait and recovered it in the form of Hp-Hb complexes (Fig. 4a).

Hp captured from sera of different donors exhibited distinct total abundances in SDS-PAGE and individual-specific oligomeric distribution patterns in native MS (Fig. 4b). Each Hp molecule is composed of an equal number of light and heavy chains linked through disulfide bonds. In this study, an *n*-mer of Hp is defined as an assembly incorporating *n* light chains and *n* heavy chains. While Individual P1 predominantly had Hp pentamers, P2 had additional Hp tetramers, and P3 carried larger-sized oligomers. Hp species were not detected at an adequate abundance in any oligomeric state in donor P4. The absence of Hp content in P4, initially identified in this SNAP-MS measurement, was later confirmed by a clinical test using the standard immuno-turbidimetry approach (Supplementary Table 2). Genetic analysis revealed P4's Hp$^0$/Hp2 genotype, which is associated with anhapto-globinemia. Hb occupancy in the Hp-Hb complexes, in response to the introduction of free Hb into the serum samples (mimicking hemolysis conditions), could be clearly revealed by native MS as well (Fig. 4c). The attachment of a linker to the bait Hb resulted in a 126 Da mass shift of Hb subunits. This mass increase serves as a mass tag that allows for the differentiation of bait Hb from plasma Hb upon releasing Hb from the bait-target complex in tandem MS (Fig. 4d). The unmodified Hb β chain was undetectable in untreated serum samples and was detected after the introduction of the hemolysis-mimicking agent, suggesting the absence of endogenous Hb in the complexes.

In the mass spectrum of a glycoprotein, the diversity in the masses of carbohydrate residues and the variations in the compositions of glycan chains inherently broaden the distribution of protein masses and result in highly convoluted signal peaks. This occurs irrespective of the resolving power of the mass spectrometers, experimental methods, or data acquisition parameters. The removal of glycans from these proteins using glycosidases is not a universally applicable solution to this issue due to the low efficiency of deglycosylation under non-denaturing conditions (Supplementary Fig. 18b). In the case of Hp, the spectra are further complicated by high heterogeneity in terms of chain lengths, oligomerization states, and binding states, in addition to glycosylation. Due to extensive signal overlap and inconsistency of mass profiles across different charge states, it becomes challenging to deconvolute the MS1 profile of heterogeneously glycosylated Hp from plasma and to unambiguously determine their masses and binding stoichiometries.

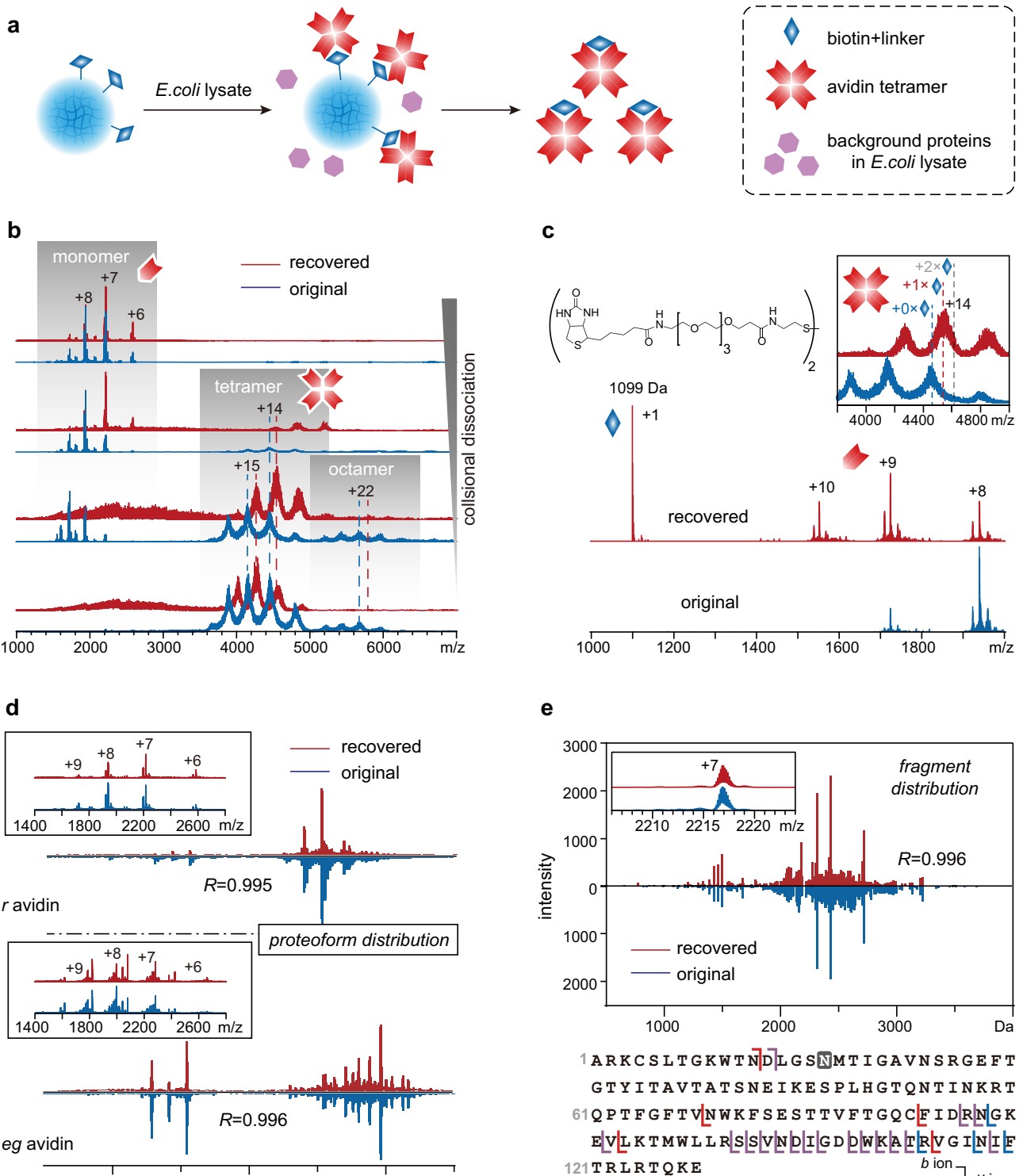

**Fig. 3 | Purification and characterization of avidin using SNAP-MS. a** Schematic diagram illustrating the purification of avidin using biotin as bait. **b** Native mass spectra of the recovered (red) and original (blue) *r* avidin arranged from bottom to top in ascending order of the in-source fragmentation settings 5, 35, 50 and 100, respectively. **c** Tandem mass spectra displaying linker-attached biotin (structure shown above) and monomeric avidin released in collisional dissociation of the avidin tetramer. The inset illustrates the mass increase of tetramers due to bait incorporation, with dashed lines indicating the theoretical masses of tetrameric avidin incorporating 0, 1 and 2 copies of linker-attached biotin. **d** Proteoform distributions of monomeric *r* avidin and *eg* avidin released from tetramers, as revealed

by their deconvoluted mass spectra. Comparisons of spectra of recovered (red) and original (blue) avidin were evaluated using the Pearson correlation coefficient (*R*). Insets show raw mass spectra of these monomers. **e** Comparison of pseudo-MS3 spectra of a representative proteoform of recovered (red) and original (blue) avidin monomers, evaluated with Pearson correlation coefficient (*R*). The Inset shows spectra of mass-selected precursor monomer ions without fragmentation. Fragmentation sites at recovered, original, and both avidin samples are indicated by red, blue and violet delimiters, respectively, on sequence shown below the spectra. Source data are provided as a Source Data file.

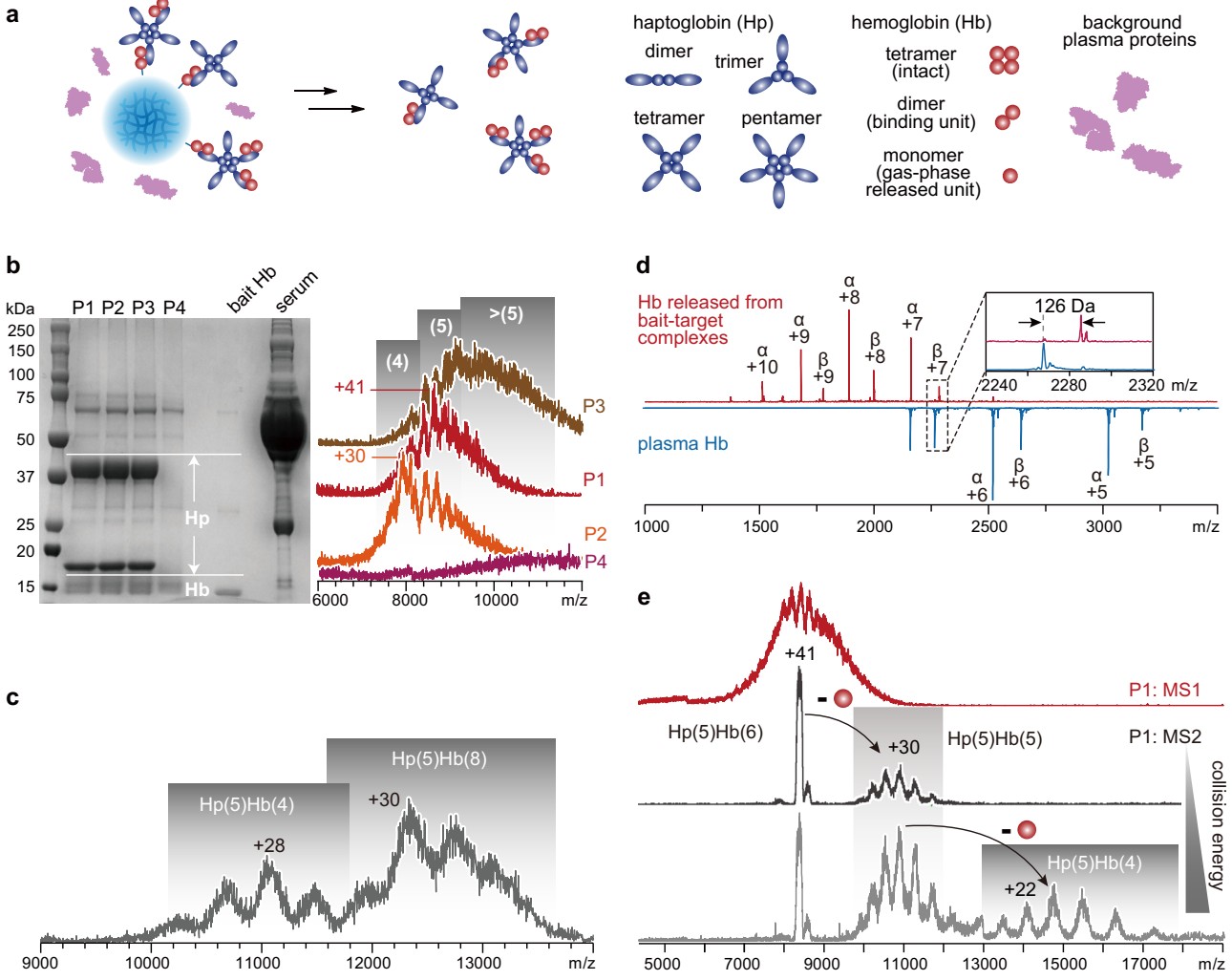

**Fig. 4 | Purification and characterization of Hp from human plasma using SNAP-MS. a** Schematic diagram illustrating the purification of Hp using Hb as bait. **b** SDS-PAGE (left) and native mass spectra (right) of Hp-Hb complexes captured from sera of four individuals (P1-P4). Pure Hb bait and untreated serum from Individual P1 were loaded in the gel as controls. Numbers in parentheses indicate oligomeric states of Hp. The SDS-PAGE experiment was independently repeated three times, yielding consistent results. **c** Native mass spectrum of Hp-Hb complexes captured from 100 μL of serum from P1, with 50 ug free Hb added. 30 mM TEAA was added in the sample solution to reduce the charge states of protein ions

for improved charge resolution. Numbers in parentheses indicate binding stoichiometries. **d** Comparison of tandem mass spectra of Hb monomers released from plasma Hb (blue) and from the bait-target complexes (red). The inset shows a magnified view of the mass increase of the Hb β subunit caused by linker attachment. **e** Tandem mass spectra showing sequential release of monomeric Hb subunit from bait-target complexes with increasing collision energy, which facilitates determination of complex masses and stoichiometries. See Supplementary Fig. 18c for a detailed algorithm. Source data are provided as a Source Data file.

In this context, the bait molecule, which binds the target during capture and remains attached to the target after the dissolving of SNAP beads, can serve dual roles as a charge remover and mass corrector. This enables accurate determination of mass and stoichiometry for the targets. According to the asymmetric charge partitioning effect in tandem MS[35–38], as mentioned previously, the ejection of an Hb bait subunit from the bait-target complex through collisional dissociation during tandem MS allows for the removal of multiple charges (Fig. 4d). This results in a significantly reduced charge density of the complementary target complex (Fig. 4e). Such charge reduction offers additional benefits for the accuracy of mass/charge determination by increasing charge resolution and tolerance of m/z reading uncertainties[31]. Furthermore, the defined mass of the bait allows for the correction of inaccurate mass determination for the target using conventional algorithms, which is due to the fact that inconsistent mass profiles of the target ions in different charge states invalidate the basic assumptions of these algorithms. By providing an additional constraint for mass determination based on the mass balance of the

dissociation reaction (*i.e.*, the total mass of bait-target complexes equals the sum of the bait mass and the target mass (Supplementary Fig. 19)), these inaccuracies can be corrected. Increasing collision energy allows for the sequential release of more than one subunit from the bait-target complex. This enables mass determination for complexes showing unresolved MS1 profiles or verification of results derived from the primary dissociation event (Fig. 4e). By leveraging the aforementioned charge reduction and "fragment complementation" effects provided by the bait molecule, SNAP-MS offers a universal means to access mass and binding information of heterogeneous glycoprotein systems, including endogenous ones from biological samples.

As previously discussed, when removing the bait from the bait-target complex for target characterization, a decision can be made between releasing a bait subunit and a target subunit as the ejected subunit during collisional dissociation. When the bait is smaller than the target protein or any subunit of the target complex, the bait is preferentially ejected, and vice versa. In the example of characterizing

the Hp(2)Ab(1) complex, an increase in collisional energy resulted solely in the sequential release of the Hb bait subunit, without compromising the integrity of the target Hp-Ab complex (Supplementary Fig. 20). This observation underscores that the integrity of the target complex can be effectively preserved by employing a bait of smaller size.

While Hp is among the abundant proteins in plasma, its highly heterogeneous nature, unlike relatively homogeneous proteins such as albumin, results in the co-existence of a broad distribution of proteoforms and oligomerization/binding states. These states exhibit a wide range of masses, which significantly dilutes the detectable signal of each proteoform. To estimate the efficiency of SNAP-MS in characterizing such a complex plasma protein system, we conducted protein quantification and sequential dilution of Hp purified from Individual P1. According to the critical dilution fold that resulted in adequate spectral quality for reliable data interpretation and BCA-based quantitation (Supplementary Fig. 21), Hp at a concentration higher than 3 ng/μL could be effectively detected by native MS. Given the heterogeneous nature of Hp, the minimum concentration of a less heterogeneous protein needed for effective characterization can be significantly lower.

### Screening optimal expression and purification strategy for targeted protein complexes and complementing cryo-EM for low-input samples

Since SNAP-MS allows characterization of the integrity, subunit composition, and topology of target complexes following rapid and efficient purification, it provides concurrent information on both the integrity and degradation of the protein complexes under investigation, which includes their dissociation and rearrangement. As a result, it offers substantial advantages for screening optimal strategies for protein expression and sample preparation, thereby enhancing the quality of data provided by conventional structural biology tools such as cryo-EM and improving their workflows. The *Thermoplasma acidophilum* 20S proteasome (20S), a 28-mer complex consisting of two outer layers of heptameric α subunits and two inner layers of heptameric β subunits (Fig. 5a), is a model system frequently used in structural biology. Native MS analysis of 20S purified from cell lysate in previous studies revealed the presence of a 14-mer of the α subunits, in addition to the intact 28-mer. Given that there is no direct contact between the two heptameric α layers in the naturally intact 20S, the presence of such a 14-mer suggests the occurrence of complex rearrangement during sample preparation prior to MS detection[43–47]. For biophysical characterization, intact complexes with higher stability that resist such rearrangement are preferred to minimize interference from rearrangement artifacts.

Here we utilized SNAP-MS to rapidly screen 20S expressed using different strategies, which resulted in varying degrees of resistance to degradation (Fig. 5a). Following the overexpression of an unmodified α subunit and a β subunit modified with a Twin-Strep-tag, and the self-assembly of α and β, we used SNAP beads conjugated with ST to purify the assembled 20S. Compared to bait-cleavable agarose beads, SNAP beads exhibited remarkably high purification efficiency (Supplementary Fig. 22). The high efficiency of the SNAP workflow allows for the purification of picomole-level 20S from less than 1 mL of *E. coli* liquid culture in a petri dish (Supplementary Fig. 23). When α and β were expressed from two separate plasmids that individually code for α and β respectively, the purified 20S particles exhibited heterogeneous compositions as characterized by native PAGE, SEC, and negative stain EM (Fig. 5b). Western blotting with anti-Strep Ab confirmed the presence of β subunits in the purification products (Fig. 5c). Complexes detected in native MS were predominantly heptamers of α subunits, a dissociation product of 20S, and 14-mers of α, a rearrangement product (Fig. 5d), suggesting low resistance to degradation of complexes expressed under this condition. MS characterization of the proteins expressed from a single plasmid that codes only for the α subunit

confirmed that these degradation products can be spontaneously assembled in the absence of the β subunit (Supplementary Fig. 24).

In contrast, when the two subunits were coded in the same plasmid for expression, native MS detected a predominance of intact 20S over rearranged subcomplexes, revealing the production of more stable intact 28-meric 20S (Fig. 5e). The activity of the purified 20S was verified with in-gel and in-solution peptidase activity assays respectively (Supplementary Fig. 25). Following condition screening by SNAP-MS, the sample with the highest abundance of the intact 20S was subjected to EM. Since the polyacrylamide polymer presents little signal under EM, after dissolving the SNAP beads that captured the target, we were able to directly transfer the dissolving product to a cryo-EM grid for the characterization of target complexes without the removal of the gel polymers. With the dissociation and rearrangement products significantly reduced, images of an increased number of intact 20S particles with adequate clarity and particle density were collected (Supplementary Fig. 26). This not only confirmed the integrity of virtually all purified 20S particles but also allowed for the reconstruction of a 3D structure of 20S with a 3.16 Å resolution (Fig. 5f). Thus, SNAP-MS provided efficient purification for target complexes from low-input cell lysate samples and rapid screening for conditions that maximize the purity of the target.

## Discussion

Reducing the reliance of biophysical characterization of protein complexes on in vitro reconstituted complexes, and enabling direct characterization of native complexes from biological samples, are crucial for accessing the structural and functional information of a wider array of protein systems in greater detail and with higher fidelity. This objective faces challenges such as reducing interference from the high background complexity of biological samples in low quantities, preserving the fragile non-covalent interactions that maintain the higher-order structures of proteins, and resolving the heterogeneity caused by diverse states of PTM, oligomerization, and binding. Affinity purification enables effective capture of target protein species, but the need to preserve the integrity of non-covalent structures prevents the recovery of captured species through elution under denaturing conditions. The approaches to isolate or enrich low-abundant proteins in biological samples using nanomaterials either are incompatible with native condition[48–50] or requires elution buffer specifically designed for preservation of non-covalent interactions[51]. As explained in the Introduction section, employment of competitive elution under native conditions creates a dilemma that limits the efficiency of either target capture or target recover. Recovering the captured proteins through cleavage of the bait-bead linker and release of the target-bait complexes provides a means to circumvent this dilemma. However, comparisons of the performance of different types of beads in purifying multiple target systems (Fig. 2 and Supplementary Fig. 21) suggest that this bait-cleavable scheme does not guarantee a high recovery rate. Factors limiting the recovery include insufficient accessibility of reagents to cleavable sites at the bead surface and non-specific adsorption of background proteins at the bead surface. In addition, since the interstitial voids between the spherical beads retain a significant quantity of proteins, an elution step cannot be omitted to recover this portion of proteins, which significantly dilutes the output sample. All these factors arise from the co-existence of the solid and solution phases in the purification system. Dissolving the bead matrix to convert the solid phase into a solution provides a solution to these problems. In our design of purification workflow using SNAP beads, dissolving the bead matrix completely eliminates the solid phase and the associated steric hindrance for cleavage reactions, surface adsorption, and interstitial voids. The combination of matrix dissolving and bait cleavage strategies, rather than bait cleavage alone, allows complete omission of the elution and significant improvement of target recovery.

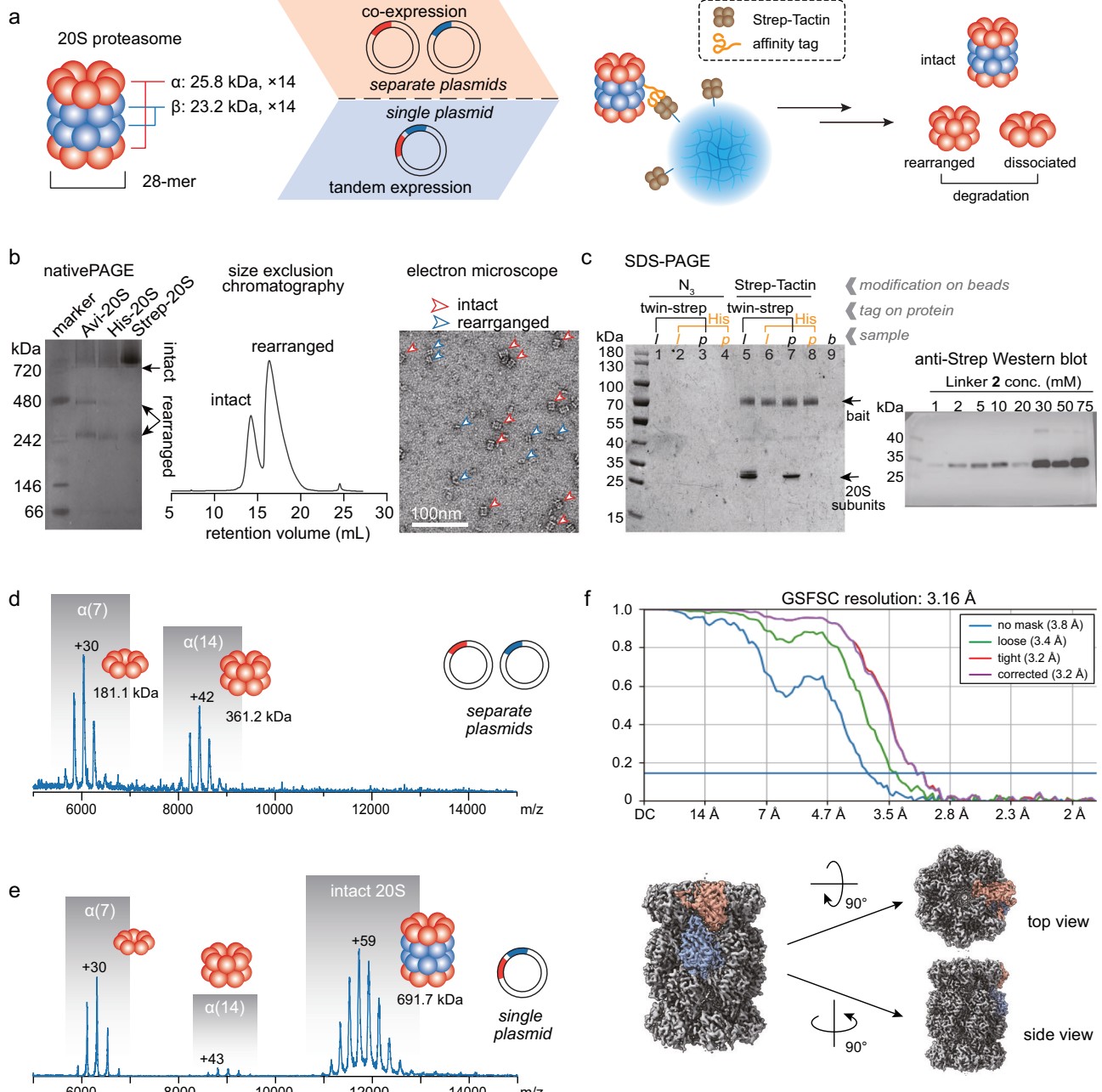

**Fig. 5 | Purification and characterization of 20S proteasome using SNAP-MS.**
**a** Schematic representation of hierarchy, purification, and rearrangement of 20S proteasome. **b** Native PAGE, SEC, and native stain EM characterization of purified 20S, with α and β subunits expressed from separate plasmids. The SDS-PAGE experiment was independently repeated three times, yielding consistent results. **c** Detection of β subunit from samples containing 20S expressed using separate plasmids. Left: SDS-PAGE of products from the purification of Twin-Strep- or His-tagged proteins using $N_3$- or ST-modified beads (*l*: cell lysate; *p*: pure 20S; *b*: blank). Right: Western blot (with anti-Strep Ab) of the 20S purified using SNAP beads synthesized with different concentrations of Linker **2**, showing the presence of β subunit. **d** Native mass spectra of purified 20S expressed from separate plasmids. **e** Native mass spectra of purified 20S expressed from a single plasmid. **f** 3D-reconstructed density map of the purified intact 20S structure at a resolution of 3.16 Å using 120 micrographs. Source data are provided as a Source Data file.

In the purification products, the detected background proteins may encompass those non-specifically bound by the bait and those adsorbed onto the bead surface. Our proteomic analysis has revealed the identities and relative quantities of these background proteins, which were captured by SNAP beads conjugated with different baits (Supplementary Fig. 27). The reproducibility of the results, derived from three measurements for each bead type, suggests that the capture of background proteins is not entirely random. The abundance distributions of these detected background proteins display bait-specific patterns, indicating that the bait identity influences the non-specific capture. The isoelectric point (PI) distributions of these

background proteins imply that there are other factors, apart from electrostatic interaction, that drive the non-specific capture. Consequently, the observed correlation between the detected quantity of background proteins and the affinity of the bait conjugated on the beads (Fig. 2) can be partially attributed to the specificity of each bait. Therefore, the enhanced flexibility in bait selection, facilitated by the SNAP-MS scheme, promotes not only the target recovery rate but also the specificity of purification.

The heterogeneity problem caused by extensive PTM often hinders interpretation of mass spectra of intact proteins from biological samples. Electrospray ionization (ESI) is the preferred method for

native MS. However, the multiple-charging nature of the electrospray protein ions, non-identical charging behaviors between different proteoforms, and distribution rules of the m/z values of the multiply charged ions often result in highly convoluted and uninterpretable mass profiles. Since accurate mass and stoichiometry determination of the analyte proteins require definite charge values of the corresponding ions in addition to their detected m/z, accurate charge determination is pivotal for spectra interpretation. Conventional algorithms based on statistics related to the m/z rule[52–63] often lead to inaccurate results due to the inconsistent mass-profile problem[28]. These algorithms rely on adequate resolution of adjacent charge states of protein ions. Algorithms developed for isotopically resolved signals[64–67] do not apply to most glycoprotein systems whose signals are not isotopically resolved due to intrinsic mass heterogeneity, regardless of the instrument resolving power. CDMS, which directly measures charge during ion detection[29,30], relies on specific model of instrument. Charge reduction of protein ions benefits charge determination in multiple aspects, including improving charge resolution and tolerance to m/z reading uncertainties[31]. However, global reduction for proteoform ensemble does not suffice for accurate charge determination of highly heterogeneously glycosylated species. The limited charge reduction approach solves these problems by isolating a subpopulation of ions in a single charge state with a narrow mass distribution, followed by producing ions in a series of lower charge states with mass profiles resembling that of the precursor ions[28]. However, this approach relies on electron-transfer, electron-capture or proton-transfer reactions, which are functions available in limited models of instruments.

The SNAP-MS workflow, which involves a cycle of bait-attachment in solution and bait-removal in the gas phase, provides a convenient approach to address the heterogeneity problem caused by extensive PTM. As detailed in the Results section, the bait in the bait-target complex, produced through the dissolution of SNAP beads, serves dual roles as a charge remover and mass corrector during the bait removal process. The ejection of the bait from the bait-target complex effectively removes charges from the target portion, leading to charge reduction of the target portion and an improvement in charge resolution. Since the reduction products are derived from the mass-selected precursor ions, the mass profiles of the product ions in the reduced charge states resemble that of the precursor charge state, thereby enhancing the accuracy of mass determination for the target. Consequently, the bait-removal operation can essentially achieve all the objectives aimed at by the limited charge reduction approach. Moreover, in the "fragment complementation" approach, the defined mass of the bait molecule provides the mass balance of the dissociation reaction it participated in as an additional constraint, further improving the accuracy of mass determination for the target (refer to the caption of Supplementary Fig. 18c for a detailed mass determination algorithm). The achievement of all these functions is facilitated by the bait attachment during the upstream purification and collisional dissociation, a function readily available in a wide array of instrument models for intact protein analysis, without the need for additional reagents or electron-or proton-manipulation functions from the instrument.

When a multi-subunit complex is employed as bait, additional signal clusters corresponding to residual subcomplexes from secondary and even tertiary dissociation events could be produced through the increase in collision energy. The masses of residual subcomplexes from subsequent dissociation events, benefiting from the further improved charge resolution, can serve as either one more constraint for mass determination or verification of the results derived from primary dissociation events. In the example illustrated in Supplementary Fig. 18, the "fragment complementation" approach verified the determined charge states of signals labeled as +i series. These signals are both the dissociation products in primary dissociation and

precursors in secondary dissociation. This approach also corrected the charge states of +i series signals by one unit. Such a charge error corresponds to a mass error of 15 kDa, corresponding to a stoichiometry error of one Hb subunit. With this improved accuracy, the stoichiometry of species giving rise to +i series signals was unambiguously identified. This information could then be used to determine the stoichiometry of the original intact complexes, whose signals are not adequately resolved for reliable mass determination.

When considering the impact of the bait's presence on the structural characterization of target proteins, three scenarios are pertinent: intact mass and stoichiometry analysis of proteins and complexes (Type 1), intact mass and top-down analysis of individual proteins (Type 2), and subunit and topology analysis of protein complexes (Type 3). For Type 1 analysis, the defined mass of the bait implies that its attachment does not influence the mass or stoichiometry analysis of the target proteins or protein complexes, thereby obviating the need for bait release. This is illustrated by three cases presented in MS1 spectra in Figs. 2g, h and 3b, where the bait is significantly larger than the target, slighter smaller than the target, and significantly smaller than the target, respectively. In Type 2 analysis, the bait can be released from the complexes through collision-based dissociation, which preferentially ejects a non-covalent subunit and does not compromise the integrity of the target protein. This is demonstrated in the examples of characterizing GFP variants after release of Ab or sdAb bait (Fig. 2g, h) and avidin monomers after release of the biotin bait (Fig. 3d, e). In Type 3 analysis, the use of a small-sized bait ensures the integrity of the target complex. Protein complexes have a tendency to eject smaller and peripheral subunits during collisional dissociation[34]. Given that the bait-target interaction is typically situated peripherally within the bait-target complex, employing a smaller bait can facilitate its ejection under collision conditions that do not disrupt the target complexes. This concept is illustrated in Supplementary Fig. 19, where Hb was used as the bait for the target complex Hp(2)Ab(1). During collisional dissociation, the bait-target complex Hp(2)Hb(4)Ab(1) sequentially released Hb subunits across a range of collision energies, all the while maintaining the integrity of the target complex. Even under extreme conditions, which may potentially trigger the release of a target subunit prior to the ejection of the bait, the stoichiometry analysis of the target complex remains unaffected due to the defined mass of the bait, which allows for accurate stoichiometric analysis regardless of the sequence of subunit ejection.

With its matrix-dissolvable and bait-cleavable attributes, SNAP-MS offers enhanced flexibility in the selection of bait-target pairs. This adaptability is advantageous for the purification of not only over-expressed protein systems from cell lysates, facilitated by the use of high-affinity purification tags, but also natural protein complexes from biological samples. In the latter case, where no tag can be introduced at the expression stage, SNAP-MS allows a broader choice of appropriate natural or engineered ligands or binding proteins as bait. When the chosen bait is identical to any endogenous species present in the biological sample, concerns regarding interference can be mitigated by mass measurement of the ejected subunits in tandem MS. As illustrated in Fig. 4d, the residual moiety of the bait-bead linker, which remains covalently linked to the bait following bead dissolving, can function as a mass tag that enables the differentiation of the bait from its endogenous counterparts. The efficacy of SNAP-MS in preserving and accurately revealing the distribution of sequence variants, PTM, and the oligomerization and binding states of target proteins, as demonstrated in this work, enables access to individual-specific proteoform distribution information. This information can elucidate the molecular mechanisms associated with genetic and health conditions.

In the context of characterizing natural Hp-related complexes from human sera, which display high heterogeneity in terms of phenotype, glycosylation, oligomerization, and Hb binding patterns[42], SNAP-MS has

proven efficient in both purification and characterization of native Hp complexes. It enables profiling of individual-specific Hp oligomerization and Hb-binding states, as well as the mass distribution that reveals carbohydrate contents. The aforementioned heterogeneity-resolving strategy, based on bait-release in tandem MS, offers a convenient approach to protein identification and stoichiometry determination in such intricate systems. In addition to its critical role in binding free Hb for tissue protection and oxidative damage prevention, Hp is implicated in the prevalence and clinical progression of numerous inflammatory diseases, including infections, atherosclerosis, and autoimmune disorders[68]. Prior studies have suggested a potential association between Hp polymorphism and diseases such as diabetes and albuminuria[69–71]. Furthermore, Hp is a potential biomarker for diseases including various forms of malignant neoplasms[72]. SNAP-MS introduces a promising approach for the analysis of Hp polymorphism, offering superior resolution for its coexisting states compared to traditional methods like native PAGE and immunoturbidimetry[73,74] and providing clinically valuable information.

In conclusion, the SNAP-MS workflow significantly enhances the efficiency of purifying protein complexes from biological samples through its bead-dissolving scheme and improves the feasibility and accuracy of structural characterization of the target proteins using native MS and the associated top-down analysis. SNAP beads are compatible with a diverse range of baits, which facilitate high-affinity purification of not only overexpressed proteins from cell lysates but also natural protein complexes from blood samples. Native MS provides multi-level structural information on the purified proteins, including sequence, PTMs, oligomerization, binding, and complex topology. It can also screen for optimal expression and preparation conditions for further structural characterization using conventional tools such as cryo-EM. Bait-release from the bait-target complexes in the workflow enables interpretation of spectra of highly heterogeneous protein systems acquired a wide range of commercial mass spectrometers. This facilitates accurate higher-order structure characterization of heterogeneously glycosylated protein systems. SNAP-MS demonstrates the potential to liberate structural characterization from reliance on in vitro reconstituted protein complexes, and to serve as a general strategy for characterizing natural protein complexes in studies employing biological samples.

# Methods
## Ethical statement
The blood samples utilized in this research were donated by four volunteers, from whom written informed consent was obtained, in accordance with Peking University People's Hospital Ethical Regulations for Research Involving Human Subjects (#2023PHB273-001).

## Materials and reagents
The following materials and reagents were used in this study:

*AAT bioquest*: TAMRA-NHS (370).

*Abbkine (China)*: HRP-goat anti-mouse IgG and anti-His-tag mouse monoclonal antibody (5C3) for Western blot of 20S.

*Aladdin Scientific (China)*: acryloyl chloride (A104614), 1-hydroxybenzotriazole monohydrate (HOBt, H106176), 4-[4-(1-hydroxyethyl)−2-methoxy-5-nitrophenoxy] butanoic acid (HMNB, H133255), 3,6,9,12-tetraoxatetradecane-1,14-diamine (T122119), and triethylamine (T431604).

*Bio-Rad*: Precision Plus Protein marker (1610374) and Strep-Tactin-HRP (1610380).

*Genescript*: Precast SDS-PAGE gel (M42015C) and loading buffer (M00138).

*IBA Lifesciences*: 1X Buffer E (2-1000-025), GFP-strep protein (2-1007-105) and Strep-Tactin XT agarose beads (2-5030-002).

*MCE*: PC-Alkyne-PEG4-NHS ester (HY-140139)

*NEB*: PNGase F (P0704S); GlycoBuffer 2 (B3704S).

*Novec*: HFE-7500.

*Promega*: trypsin/Lys-C Mix (V5073).

*Sigma-Aldrich*: acrylamide (A9099), 2-aminoethylmethacrylamide hydrochloride (900652), ammonium acetate (A2706), ammonium bicarbonate (5.33005), ammonium persulfate (APS, 248614), anti-GFP monoclonal IgG antibody produced in mouse (G6539), avidin proteins (*r*Avidin, A8706; and *eg*Avidin A9275), copper(II) sulfate pentahydrate (CuSO₄; 203165), cystamine (C8707), DL-Dithiothreitol (DTT, 43815), 1H,1H,2H,2H-Perfluoro-1-octanol (PFO, 370533), haptoglobin proteins (whole plasma Hp, H3536; Hp 1-1, H0138; Hp 2-2, SRP6508), hemoglobin (Hb, H7379), hexafluorophosphate benzotriazole tetramethyl uronium (HBTU, 12804), Iodoacetamide (IAA, I6125), maleimide-PEG4-DBCO (760676), N,N'-Bis(acryloyl) cystamine (A4929), N,N'-methylene diacrylamide (V900301), N,N,N',N'',N''-pentamethyldiethylene-triamine (PMDETA; 369497), N,N,N',N'-tetramethyl ethylenediamine (TEMED, T9281), N-ethylmaleimide (04259), NHS-PEG4-DBCO (764019), NHS-SS-biotin (21441), NHS-SS-DBCO (761532), sodium L-ascorbate(A4034), streptavidin (S4762), SPAN 80 (S6760), Tween 20 (P7949), and urea (U5378).

*Thermo Fisher Scientific*: M270 amine Dynabeads (14307D), M280 streptavidin Dynabeads (11206D), NHS-PEG4-azide (26130), NHS-PEG4-biotin (21362), PBS 1X buffer (pH = 7.4; 10010023), and tris(2-carboxyethyl)phosphine (TCEP) solution (77720).

*Tokyo Chemical Industry*: Isopropyl palmitate (P0005).

*Tong Guang Fine Chemicals (China)*: DCM (102045-2), DMSO (D103273), methanol (104028) and THF (102045-2).

## Synthesis of cleavable linkers
*Chemically cleavable Linker 1* (Supplementary Fig. 1): A cystamine solution (12.5 g, 0.08 mol) and triethylamine (5.7 g, 0.056 mol) were dissolved in methanol (200 mL) and cooled to 0 °C using an ice bath. Acrylate chloride (7.0 g, 0.06 mol), dissolved in methanol (100 mL), was added dropwise over a period of 30 min while stirring. The reaction mixture was then allowed to reach room temperature and stirred overnight. The precipitated solid was collected via filtration and washed with tetrahydrofuran (THF). The solvent was removed using rotary evaporation, and the crude product was purified through column chromatography using gradient elution (gradually increasing the MeOH:DCM ratio from 0 to 1:20). The product, a white solid, was recovered (3.5 g, 26% yield) and confirmed by HNMR and MS (Supplementary Fig. 2). ¹H-NMR (400 MHz, D6-DMSO): δ 8.39 (s, 3H), 8.26 (s, 1H), 5.69 (s,1H), 5.34 (s, 1H), 3.38 (s, 4H), 3.03-2.99 (dd, 4H), 2.84 (t, 2H), 1.83 (s, 3H).

*Intermediate for photo-cleavable Linker 2* (Supplementary Fig. 1): A solution of 4-[4-(1-hydroxyethyl)−2-methoxy-5-nitrophenoxy] butanoic acid (50.0 mg, 0.16 mmol), 3,6,9,12-tetraoxatetradecane-1,14-diamine (18.6 mg, 0.08 mmol), and triethylamine (40.0 mg, 0.40 mmol) was prepared in dichloromethane (10 mL) and cooled to 0 °C in an ice bath. HBTU (74.9 mg, 0.20 mmol) and HOBt (5.4 mg, 0.04 mmol) were added, and the reaction mixture was allowed to reach room temperature and stirred overnight. The reaction was protected from light by wrapping the flask with aluminum foil. The reaction was quenched with water (10 mL), and the organic layer was separated. The aqueous layer was extracted with dichloromethane (2 ×10 mL), and the combined organic extracts were washed with brine (10 mL) and dried over magnesium sulfate. The product was purified through column chromatography using gradient elution (gradually increasing the MeOH:DCM ratio from 0 to 1:15). The product, a yellow oil, was recovered (51.4 mg, 80% yield) and confirmed by HNMR and MS (Supplementary Fig. 3). ¹H-NMR (400 MHz, CDCl₃): δ 7.56 (s, 1H), 7.33 (s, 2H), 7.28 (s, 2H), 6.50 (s,2H), 5.58-5.53 (q, 2H), 4.12-4.10 (t, 4H), 3.98 (s, 6H), 3.61 (s, 4H), 3.58 (s, 6H), 3.51 (s, 10H), 2.44-2.42 (d, 4H), 2.22-2.19 (t, 4H), 1.57-1.55 (d, 6H).

*Photo-cleavable Linker 2* (Supplementary Fig. 1): A solution of the intermediate (340 mg, 1.0 mmol) and triethylamine (255 mg, 2.5 mmol) was prepared in dichloromethane (30 mL) and cooled to 0 °C in an ice

bath. Acryloyl chloride (133 mg, 1.2 mmol) was added dropwise via a syringe, and the reaction mixture was allowed to reach room temperature and stirred overnight. The reaction was protected from light by wrapping the flask with aluminum foil. The reaction was quenched with water (30 mL), and the organic layer was separated. The aqueous layer was extracted with dichloromethane (2 ×30 mL), and the combined organic extracts were washed with a saturated sodium bicarbonate solution (3 ×30 mL) and dried over magnesium sulfate. The product was purified through column chromatography using gradient elution (gradually increasing the MeOH:DCM ratio from 0 to 1:15). The product, a yellow oil, was recovered (292 mg, 85% yield) and confirmed by HNMR and MS (Supplementary Fig. 4). $^1$H-NMR (400 MHz, CDCl$_3$): δ 7.58 (s, 2H), 7.26 (s, 2H), 7.00 (s, 2H), 6.55-6.50 (q,2H), 6.45-6.41 (d, 2H), 6.20-6.13 (dd, 2H), 5.88-5.86 (s, 6H), 4.12-4.09 (t, 4H), 3.93 (s, 6H), 3.64-3.45 (m, 20H), 2.43-2.40 (t, 4H), 2.21-2.17 (t, 4H), 1.66-1.65 (d, 6H).

## MS characterization of the synthesized molecules

The synthesized linkers and their key intermediates were characterized using intact mass (MS1) and tandem mass spectrometry (MS2 and MS3). These measurements were performed using a Cell miniature MS system (PURSPEC Technologies, China), equipped with a standard electrospray ionization (ESI) source. The original solvent (DMSO) was replaced with a 50:50 (v/v) mixture of DMSO and methanol. The analytes were then electrosprayed from a standard tapered borosilicate capillary. In the MS1 mode, the settings were as follows: polarity was set to positive, spray voltage was set to 3500 V, detection range was set from 50 to 1,000 m/z, inj_size voltage was set to 40, and inj_lmco m/z parameter was set to 120. Tandem MS measurements were conducted by selecting precursor ions, followed by collision-induced dissociation (CID). The collisional energy was adjusted to generate fragments while maintaining adequate signal intensities. The CID q parameter was set at 0.25, and the isolation swift amplification parameters 1 and 2 were set at 2.0 and 3.5, respectively.

## Casting of chemically dissolvable SNAP beads and side-chain cleavable hydrogel beads

A 5× stock solution of N,N'-Bis(acryloyl)cystamine was prepared by dissolving N,N'-Bis(acryloyl)cystamine (0.75 g, 2.8 mmol) in a 30% w/w acrylamide solution (100 mL). Two types of hydrogel precursor solutions were subsequently prepared:

(1) for SNAP beads: a mixture was prepared using the acrylamide solution, N,N'-Bis(acryloyl)cystamine stock solution (2 mL), ammonium persulfate (APS, 0.5% w/w, 0.5 mL), and *Linker 1* (30 mM, 0.5 mL).

(2) for side-chain cleavable non-dissolvable hydrogel beads: a mixture was prepared using the acrylamide solution (6% w/w, 10 mL), N,N'-methylene diacrylamide (BIS, 0.2% w/w, 0.2 mL), APS (0.5% w/w, 0.5 mL), and Linker 1 (30 mM, 0.5 mL).

All aqueous solutions were filtered through 0.22 μm syringe filters prior to use. The oil phase was prepared by mixing HFE-7500 (100 mL) with EA surfactant (1% w/w, 1 mL) and TEMED (0.8% w/w, 0.8 mL). The hydrogel microbeads were fabricated by emulsifying the precursor solutions into the oil phase using a microfluidic device. The precursor solution was emulsified with microfluidic chips containing cross junctions at flow rates of 400 μL/h for the aqueous phase and 1200 μL/h for the oil phase. The generated droplets were collected in a 1.5 mL tube containing 400 μL of mineral oil to prevent solution evaporation. Polymerization occurred at room temperature overnight. Following the removal of the mineral oil, HFE-7500, 100 μL PFO, and 500 μL PBS buffer containing 0.1% Tween 20 were sequentially added. The beads were collected by vortexing for 10 sec and centrifuging at 1000 × g. The PFO phase was removed, and the bead suspension was transferred into a 1.5 mL tube and washed with 500 μL PBS buffer three times. The bead suspension was stored at 4 °C for later use.

## Casting of photo-dissolvable SNAP beads

A hydrogel precursor solution was prepared by combining acrylamide (16% w/w, 10 mL), photo-cleavable Linker 2 (10 mM, 0.5 mL), APS (0.5% w/w, 0.5 mL), and 2-aminoethylmethacrylamide hydrochloride (30 mM, 0.5 mL). All aqueous solutions were pre-filtered using 0.22 μm syringe filters. The oil phase was formulated by blending isopropyl palmitate (100 mL) with Abil em180 surfactant (7% w/w, 7 mL). The hydrogel precursor solution (100 μL) was emulsified using a MiCA[32] device. The resulting emulsion was centrifuged at 15000 × g for 5 min, and the collected droplets were allowed to stand for 1 h to stabilize the emulsion. Polymerization was initiated by adding TEMED to the supernatant oil until it reached a final concentration of approximately 0.5% (v/v). The mixture was then heated at 65 °C for 1 hour. Following the polymerization, the supernatant oil was removed, and the hydrogel beads were washed with hexane containing SPAN-80 (1% w/w) and resuspended by pipetting. This hexane washing step was repeated twice, after which the hexane was removed. The hydrogel beads were subsequently washed with a PBS buffer containing Tween 20 (0.1% w/w) by vortexing for 10 sec and collected by centrifugation at 5000 × g for 1 min. The hydrogel bead suspension was transferred to a 1.5 mL tube and washed three times with PBS buffer (500 μL). The final hydrogel bead suspension was stored at 4 °C for later use.

## Modification of bait protein with DBCO

For streptavidin (SA), Strep-Tactin (ST), single domain antibody (sdAb) and hemoglobin (Hb): 100 μg SA, ST, sdAb or 1 mg Hb was subjected to two washes with PBS buffer (400 μL) using a 30 kDa (for SA and ST) or 10 kDa (for sdAb and Hb) centrifugal filter. After being transferred to a 1.5 mL tube, the washed bait protein (BP) solution was incubated with either NHS-PEG4-DBCO solution (5 μL, 1 mM; Supplementary Fig. 6a, Step a) or NHS-SS-DBCO solution (5 μL, 1 mM; Supplementary Fig. 6a, Step b) in a thermomixer at 4 °C with a stirring speed of 300 rpm for a duration of 2 h. The modified BP (BP-DBCO or BP-SS-DBCO) was subsequently washed with PBS buffer using a 30 kDa (for SA and ST) or 10 kDa (for sdAb and Hb) centrifugal filter for the subsequent conjugation with beads.

*For antibody (Ab):* 50 μg of Ab was washed twice with PBS buffer (500 μL) using a 30 kDa centrifugal filter unit. The washed Ab solution was then transferred to a 1.5 mL tube, and TCEP (15 μL, 100 μM) was added. This mixture was incubated in a thermomixer at room temperature with a stirring speed of 300 rpm for 30 min (Supplementary Fig. 6b, Step a1). The disulfide-reduced Ab was subsequently washed twice with PBS buffer (500 μL) using a 30 kDa centrifugal filter unit. The Ab solution was then incubated with maleimide-PEG4-DBCO (30 μL, 100 μM) in a thermomixer at room temperature and 300 rpm for 30 min to produce AB-DBCO (Supplementary Fig. 6b, Step a2).

50 μg of Ab was washed twice with PBS buffer (500 μL) using a 30 kDa centrifugal filter unit. After being transferred to a 1.5 mL tube, the washed Ab solution was either incubated with either NHS-SS-DBCO (10 μL, 100 μM) (Supplementary Fig. 6b, Step b) or NHS-SS-biotin (10 μL, 100 μM) (Supplementary Fig. 6b, Step c) in a thermomixer at 4 °C with a stirring speed of 300 rpm for a duration of 2 h to produce AB-SS-DBCO or Ab-SS-biotin, respectively.

The modified Ab was washed with PBS buffer (500 μL) using a 30 kDa centrifugal filter for the subsequent conjugation with beads.

## Particle size analysis of SNAP beads

A hydrogel bead suspension was prepared on a glass slide for microscopic examination. The sample was illuminated under a 4x objective lens using a fluorescence microscope (Nikon Eclipse Ti) equipped with a 532 nm filter. After acquisition of digital images, the diameters of the particles were analyzed using the ImageJ software.

## Surface modification of beads for bait-conjugation

For chemically dissolvable SNAP beads: the NHS-PEG4-azide or NHS-PEG4-biotin linker was dissolved in DMSO to a concentration of 100 mM. A 100 μL aliquot of the SNAP bead suspension was transferred into a 1.5 mL centrifuge tube, to which 30 μL of the linker solution was added. The mixture was incubated under mild rotation at 4 °C for 2 h, after which the beads were collected using a mini centrifuge (Supplementary Fig. 7a, Step a1/b1; and 7b). The supernatant was removed, and 500 μL of fresh PBS buffer containing 0.1% Tween 20 was added to resuspend and wash the beads. This washing step was repeated five times to remove any unreacted linkers. The modified bead suspension was then stored at 4 °C for later use.

The same procedure was followed to modify SNAP beads with NHS-TAMRA (Fig. 1c), or to modify agarose amino beads with the NHS-PEG4-azide linker (Supplementary Fig. 7c, Steps a1/b1).

For photo-dissolvable SNAP beads: the PC-Alkyne-PEG4-NHS ester linker was first dissolved in acetonitrile to achieve a concentration of 100 mM. Subsequently, SA (100 μg) was washed twice using a 10 kDa centrifugal filter unit with 400 μL of PBS buffer. The washed SA was then transferred to a 1.5 mL tube, to which 5 μL of 1 mM PC-Alkyne-PEG4-NHS ester was added. This mixture was incubated at 4 °C and 300 rpm for 2 h in a thermomixer. The resulting modified SA was washed again with 400 μL of PBS buffer using a 10 kDa centrifugal filter unit. For the preparation of the catalytic mixture, 10 μL of 100 mM $CuSO_4$ solution was combined with 0.25 μL of PMDETA. This catalytic mixture was then introduced to the previously modified SA and combined with 100 μL of azido-modified SNAP bead suspension. To this suspension, 2 μL of the catalytic mixture and 2 μL of 50 mg/mL sodium L-ascorbate solution were added. The entire assembly was incubated under mild rotation at room temperature for 10 min. Finally, the beads were washed twice with 500 μL of PBS buffer (containing 0.1 % Tween 20) and stored at 4 °C for future use.

For magnetic amino beads: a 200 μL aliquot of the magnetic bead suspension was washed with PBS containing 0.1% Tween 20. Then, 30 μL of the NHS-PEG4-azide linker solution (100 mM in DMSO) was added, and the mixture was incubated under mild rotation at 4 °C for 2 hours (Supplementary Fig. 7d, Step b1). The beads were recovered using a magnet stand. After removing the supernatant, 500 μL of fresh PBS buffer containing 0.1% Tween 20 was added to resuspend and wash the beads. This washing step was repeated five times to remove any unreacted linkers. The modified bead suspension was then stored at 4 °C for later use.

## Conjugation of beads with bait protein

For SNAP and agarose beads: the protein modified with DBCO (BP-DBCO, Ab-DBCO, or BP-SS-DBCO) was incubated with a 20 μL suspension of azido-modified SNAP beads (Supplementary Fig. 7a, Steps a2 and b2) or agarose beads (Supplementary Fig. 7c, Steps a2 and b2) or 100 μL suspension of azido- or SA-modified magnetic beads (Supplementary Fig. 7d, Steps a and b2) under mild rotation at 4 °C for 8 h. The beads were then washed twice with PBS buffer (containing 0.1% Tween 20) and stored at 4 °C for future use.

## Measurement of Zeta Potential

The zeta potential of SNAP beads was measured using a Zetasizer Nano ZS90 Zeta Potential Analyzer (Malvern Panalytical). A 1 mL suspension of SNAP beads was washed with 1 mL water buffer for three times, followed by removal of supernatant. The washed sample was subsequently transferred into a Disposable Folded Capillary Cell (DTS1070) for analysis. Water was used as the dispersant at a temperature of 25 °C, and the Smoluchowski model was applied as the F(ka) model with a value of 1.5.

## Modification of GFP with biotin (for evaluation of recovery rate)

GFP (50 μg) was washed twice with PBS buffer (400 μL) using a 30 kDa centrifugal filter. The washed protein solution was then transferred to a 1.5 mL tube, and biotin-PEG4-NHS solution (1 μL, 340 μM) was added. This mixture was incubated in a thermomixer at 4 °C with a stirring speed of 300 rpm for 2 h (Supplementary Fig. 6c). The modified protein (GFP-biotin) was subsequently washed with PBS buffer using a 30 kDa centrifugal filter.

## Purification and recovery of target protein

A 100 μL aliquot of the supernatant from either the lysate or serum, with scavenged disulfide reactive compounds, was incubated with a 10 μL suspension of modified microbeads. This mixture was rotated gently at 4 °C for 1 h. The beads were collected by centrifugation and washed three times with 200 μL PBS buffer. The supernatant was then removed. The hydrogel beads were dissolved at room temperature either by incubating with 5 mM TCEP (final concentration) and rotating for 10 min (for the chemically dissolvable beads), or by exposure to 365 nm UV light for 30 sec (for the photo-dissolvable beads).

## Dynamic light scattering (DLS)

Following beads dissolving, the solution was analyzed using DAWN HELEOS II (Wyatt) for dynamic light scattering. This analysis followed a standard protocol for measuring particle size distribution.

## Scavenging disulfide reactive compounds from cell lysate

N-ethylmaleimide (100 mM in DMSO) was added to the cell lysate to achieve a final concentration of 1 mM. This was followed by incubation at 4 °C for 10 min. The treated cell lysate was then centrifuged at $14,000 \times g$, and the supernatant was collected for further analysis.

## Preparation of cell lysate containing GFP

The genes encoding GFP were sub-cloned into the pEGFP cloning vector. The A206K mutation was introduced using a Q5 Site-Directed Mutagenesis Kit (NEB), following the standard protocol. The vectors were then transformed into BL21(DE3) competent cells. The transformed cells were inoculated into LB medium and cultured at 37 °C until the A600 reached between 0.6 and 1.0. Protein expression was induced by adding 0.5 mM IPTG (isopropyl-1-thio-β-D-galactopyrano-side) and incubating for an additional 4 h. The cells were then resuspended in ice-cold PBS buffer. The suspension was subjected to French press, and the crude lysate was centrifuged using an Avanti J-26S XP centrifuge (Beckman Colter) at 18,000 rpm at 4 °C for 1 h.

## Sodium dodecyl-sulfate polyacrylamide gel electrophoresis (SDS-PAGE).

A volume of 10 μL of the sample was combined with SDS-PAGE 6X gel loading buffer at a 5:1 ratio. The mixture was then subjected to heat treatment at 95 °C for 5 min to ensure proper denaturation of proteins. Subsequently, the denatured protein samples and the molecular weight markers were respectively loaded into the wells of a precast gel. The gel electrophoresis was conducted under constant voltage at 140 V for 1 h. Upon completion of the electrophoresis, protein staining was carried out using the eStain® L1 protein staining system (Genescript) in accordance with the standard staining protocol, which encompasses two staining and two de-staining cycles.

## Label-free quantitative bottom-up proteomics

Three samples were analyzed for data presented in Fig. 2b, and three samples were analyzed for Figs. 2e and 2f. Three technical replicates were analyzed for each sample. Each protein sample (10 μL) was dissolved in an 8 M urea/50 mM ammonium bicarbonate solution and incubated with dithiothreitol (10 mM) at 37 °C for 1 h. The sample was subsequently alkylated with iodoacetamide (20 mM) at 37 °C for 30 min in the dark. For digestion, the sample was diluted with a 50 mM

ammonium bicarbonate solution to reduce the urea concentration to less than 2 M. It was then incubated with a trypsin/lys-C mixture (w:w = 1:50) at 37 °C for 16 h. The digestion process was quenched with formic acid (5% v/v), and the resulting peptides were desalted using a C18 column. The peptides were eluted with acetonitrile (50% v/v) containing formic acid (0.1% v/v) and subsequently lyophilized. If the quantity of the sample varied, a BSA solution was added to adjust the sample quantity to a consistent level.

The lyophilized tryptic peptides were reconstituted with formic acid (10% v/v) and injected into an LC-MS/MS system. This system consisted of either an Orbitrap Eclipse Tribrid mass spectrometer (Thermo Fisher Scientific) coupled to an EASY-nLC 1200 nano-LC system (for data presented in Fig. 2b), or a TimsTOF Pro mass spectrometer (Bruker) coupled to a NanoElute nano-LC system (for data presented in Figs. 2e and 2f). The mobile phase was composed of water with formic acid (0.1% v/v) as solvent A and acetonitrile with formic acid (0.1% v/v) as solvent B. The peptides were trapped on a C18 trap column at a flow rate of 10 μL/min for 5 min with 3% solvent B prior to separation with EASY-nLC 1200. For the Orbitrap system, all samples were then separated on a C18 analytical column at a passive split flow of 300 nL/min for 95 min; the separation involved a linear gradient of 8–50% solvent B over 51 min, followed by 50–99% solvent B over 14 min, and 99% solvent B for 15 min. For the TimsTOF system, all samples were separated on a C18 analytical column at a passive split flow of 300 nL/min for 65 min; the separation involved a linear gradient of 5–30% solvent B over 45 min, followed by 30–37% solvent B over 5 min, and 80% solvent B for 10 min.

The raw LC-MS/MS data acquired with the Orbitrap and TimsTOF systems were processed with Proteome Discoverer v2.4.1.15 (Thermo Scientific) and PEAKS Studio X Pro v10.6 (Bioinformatics Solutions) respectively following the standard sequence database search workflow. The data were searched against a merged proteome database containing the Escherichia coli (strain B / BL21-DE3) proteome (https://www.uniprot.org/uniprotkb?query=%28taxonomy_id%3A469008%29) and GFP (P42212) from the Universal Protein Resource (UniProt). The search strategy was set as follows. A minimum length of 6 amino acids was required for protein detection. The maximum number of missed cleavages was set as 2. A false discovery rate (FDR) < 1% was set to filter peptides and peptide spectral matches. For data acquired with Orbitrap, the precursor mass tolerance was set to 10 ppm and the fragment mass tolerance was set to 0.02 Da; for data acquired with TimsTOF, the precursor mass tolerance was set to 20 ppm and the fragment mass tolerance was set to 0.05 Da.

### Preparation of cell lysate containing 20S proteasome
The genes encoding the α and β subunits of the *T. acidophilum* 20S proteasome were sub-cloned into either RSFDuet1 and pET21a plasmids, respectively, or into a single pACYCDuet1 plasmid. These constructs were tagged at the C-termini with either a Twin-Strep-tag or a His tag. The *T. acidophilum* 20S proteasome constructs were co-transformed into BL21(DE3) competent cells. The transformed cells were inoculated into LB medium and cultured at 37 °C until the A600 reached between 0.6 and 1.0. Protein expression was induced by adding 0.5 mM IPTG (isopropyl-1-thio-β-D-galactopyranoside) and incubating for an additional 4 h. The cells were then resuspended in an ice-cold lysis buffer containing 50 mM Tris-HCl (pH 8.0), 300 mM NaCl, 10% Glycerol, 1 mM EDTA, and 1 mM phenylmethanesulfonylfluoride. The suspension was subjected to French press, and the crude lysate was centrifuged at 18,000 rpm at 4 °C for 1 h.

### Purification of 20S proteasome using conventional beads
The supernatant, which contained the Strep-tagged *T. acidophilum* 20S proteasome, was collected through a 0.22 μm filter (Millipore) and loaded onto a column packed with 1 mL of Strep-Tactin Sepharose resin (IBA Life Science). The column was washed with 50 mL of washing

buffer containing 50 mM Tris-HCl (pH 8.0), 300 mM NaCl, 10% Glycerol, 1 mM DTT, and 2 mM EDTA. The protein solutions were pooled together and concentrated using a 100 kDa ultrafiltration tube (Amicon). The solution was then injected into a size-exclusion column (Superose 6 Increase 10/300 GL, GE Healthcare), and the separation was performed using a gel filtration buffer containing 20 mM Tris-HCl (pH 8.0), 150 mM NaCl, 10% Glycerol, 1 mM DTT, and 2 mM EDTA. The protein solution was concentrated in an ultrafiltration tube to a concentration of 4-6 mg/ml, and stored at −80 °C. All the above steps were executed at 4 °C.

### Preparation of human serum
Blood samples were obtained from four donors, none of whom had any previously diagnosed health conditions relevant to this study. Informed written consent was obtained from each donor. Fresh whole blood samples were incubated sequentially at 37 °C for 1 h and then at 4 °C for 4 h to facilitate fibrinogen depletion. The samples were subsequently centrifuged at 1500 × g for 10 min to remove the blood cells. The resulting sera were stored at −20 °C for future use. Before protein purification, each serum sample was treated with 100 mM N-ethylmaleimide in DMSO to achieve a final concentration of 1 mM. This was followed by incubation at 4 °C for 10 min. The treated serum was then centrifuged at 14000 × g, and the supernatant was collected for further analysis.

### Liquid chromatography (LC)
Size-Exclusion Chromatography (SEC): SEC measurements, excluding those of 20S proteasomes, were conducted using a Vanquish UHPLC system (Thermo Fisher Scientific) equipped with a 300 mm MAbPac™ SEC-1 column (Thermo Fisher Scientific). The mobile phase consisted of 150 mM ammonium acetate, and the flow rate was set at 0.1 mL/min (for Hp 1-1) or 0.2 mL/min (for other samples). Detection was achieved through the absorbance of UV light at 254 nm, 260 nm, 280 nm, and 488 nm. For the SEC analysis of 20S proteasomes, an ÄKTA pure protein purification system equipped with a Superose 6 Increase column (GE Healthcare) was used under similar parameters.

Weak Cation Exchange Chromatography (WCX): WCX measurements were performed using a Vanquish UHPLC system equipped with a 50 mm ProPac™ Elite WCX column (Thermo Fisher Scientific). The flow rate was set at 0.1 mL/min. Mobile phase A consisted of a 5 mM ammonium acetate solution containing 1 mM acetic acid, while mobile phase B was composed of 150 mM ammonium acetate containing 1 mM ammonia. The elution gradient was established by increasing the percentage of mobile phase B from 0 to 100% over a period from 4 min to 8 min.

Specific methods employed for samples characterized in this work are detailed in Supplementary Table 1.

### Native mass spectrometry (MS)
For data presented in Figs. 2–5, three, ten, eight and two samples were analyzed respectively. The buffer/salt solution of all samples was replaced with a 150 mM ammonium acetate solution using 30 kDa centrifugal filters (Amicon). Avidin, Hp and 20S samples characterized by native MS were purified with the chemically dissolvable SNAP beads. Native MS measurements were conducted with an Orbitrap Q Exactive UHMR mass spectrometer (Thermo Fisher Scientific) equipped with a commercial static nanoESI ion source. In MS1 mode, the settings were as follows: resolution – 50,000, desolvation voltage – −10 V, in-source fragmentation (SF) – 5–200 (higher SF values were used to eliminate polymer interference or to release subunits from the complexes), and detection range – 1000–15,000 or 1000–10,000 m/z. Higher-energy collisional dissociation (HCD) was employed to fragment the mass-selected precursor ions in tandem MS (MS2). The settings for this were: resolution – 25,000 or 50,000, isolation window – 5–250 m/z (narrow windows were used to isolate individual

proteoforms), and NCE – adjusted to release the fragments while maintaining sufficient signal intensities. Pseudo-MS3 was carried out by HCD of the mass-selected species that had been released by high SF.

Online SEC-MS measurements were performed using the Vanquish UPLC system coupled with the UHMR mass spectrometer. A 300 mm MAbPac™ SEC-1 column (Thermo Fisher Scientific) was used for separation, with the flow rate set at 0.1 mL/min.

Data were analyzed with Xcalibur (Thermo Scientific). Isotopically resolved mass spectra were deconvoluted with Biopharma Finder (Thermo Fisher Scientific). For a detailed description of the tandem-MS-based "fragment complementation" algorithm used for mass determination of heterogeneous glycoproteins, please refer to the caption of Supplementary Fig. 18. Theoretical and measured masses of proteins or complexes are compiled in Supplementary Table 3. Raw data from all native MS measurements are available in Supplementary Data 1.

## Native PAGE
10 μL of each purified 20S proteasome sample containing 20S tagged with Avi, His, or Strep were loaded onto a NativePAGE™ Novex 3-12% Bis-Tris Gel (Thermo Fisher). The electrophoresis was operated in Light Blue Cathode buffer at 150 V for 1 h, followed by an additional 30 min at 200 V. The gel was stained with Coomassie R-250, following the protocol provided in the vendor's manual.

## In-gel proteasome activity assay
The in-gel proteasome activity assay was performed by loading 10 μL of each sample onto a 4% native gel. The samples were His-tagged 20S purified with conventional solid beads, Avi-tagged 20S purified with SNAP beads, the dissolving products of SNAP beads, and untreated cell lysate containing 20S. The gel was placed in an ice bath and underwent electrophoresis in running buffer (100 mM Tris base, 100 mM Boric acid, 1 mM EDTA, 2.5 mM MgCl₂, 0.5 mM ATP, 0.5 mM DTT) for 3.5 to 4 h at 130 V. To detect proteasome activity, the gel was reacted in the reaction buffer with the Amplite® Fluorimetric Proteasome 20S Activity Assay Kit (AAT Bioquest) for 60 min at 37 °C. Imaging was subsequently performed at a wavelength of 488 nm.

## Negative-stain electron microscopy (EM)
Approximately 4 μL of the mixture, containing the recovered protein and dissolved microbeads, was loaded onto a glow-discharged ultra-thin film-coated grid (Electron Microscopy China) and left for 1 min. The grid was then blotted using filter paper and subsequently immersed twice in a 40 μL droplet of a staining solution containing 2% uranyl acetate. After the solution was removed from the grid using filter paper, the grid was immersed in another stain droplet for 1 min. Following an air-drying period of approximately 3 min, the grids were transferred to a FEI Tecnai Spirit Transmission Electron Microscope (Thermo Fisher Scientific) for data collection.

## Cryogenic scanning electron microscopy (cryo-SEM)
SNAP beads in the form of hydrogel microspheres with various modifications were submerged in PBS buffer. A 10 μL aliquot of the resuspended hydrogel microspheres was pipetted from the bead suspension. The original solution of hydrogel microspheres was diluted with PBS buffer at ratios of 1:0, 1:1, and 1:2, mixed thoroughly, and transferred into sample tubes for subsequent use. Upon stabilization of instrument, the hydrogel microspheres were deposited onto the surface of a standard electron microscope grid and allowed to stand at room temperature until all surface moisture had evaporated. The carrier network was added to the snap ring of the sample holder, then secured with the O-ring. The sample loading snap ring was then screwed in place to prepare for sample loading. The morphology and structure of the hydrogel microspheres were characterized using an

FEI Helios Nanolab G3 DualBeam FIB-SEM platform (Thermo Fisher Scientific).

## Cryogenic electron microscopy (cryo-EM)
An aliquot (c.a. 4 μL) of the recovered protein-microbeads mixture was applied to a glow-discharged Quantifoil Cu 1.2/1.3, 400 mesh graphene-oxide coated grid. The glow-discharge process was performed with medium force for 10 sec after vacuum pumping for 2 min. The grids were blotted using 55 mm filter papers (TED PELLA, INC.) for 2.0-3.5 sec at 10 °C with 100% humidity and then flash-frozen in liquid ethane using a FEI Vitrobot Marked IV (Thermo Fisher Scientific). The frozen grids were transferred to a FEI Talos Arctica Cryo-TEM (Thermo Fisher Scientific) equipped with a Gatan K2 camera for data collection. The Cryo-TEM was operated at 200 kV. Each stack was exposed with a total dose of 50 e−/Å2 and fractioned into 32 frames in a dose-dependent manner. The EM movie stacks were recorded at a magnification of 45,000x, yielding a pixel size of 0.94 Å, with a varied defocus ranging from −1.5 μm to −2.5 μm. For the 20S proteasome captured using SNAP, we collected 120 micrographs using SerialEM software. Following non-uniform refinement with a D7 symmetry performed in CryoSPARC v3.1.3, we refined a final electron density map at 3.16 Å resolution based on the criterion of Fourier shell correlation (FSC) = 0.143.

## Reporting summary
Further information on research design is available in the Nature Portfolio Reporting Summary linked to this article.

## Data availability
The raw files of proteomics data generated in this study have been deposited in the ProteomeXchange Consortium (https://proteomecentral.proteomexchange.org) via the MassIVE partner repository (https://massive.ucsd.edu/ProteoSAFe/static/massive.jsp) with the identifier PXD054976 (in ProteomeXchange) and MSV000095635 (in MassIVE; last accessed on Aug 18th, 2024). The cryo-EM density maps have been deposited in the RCSB Protein Data Bank (PDB; https://www.rcsb.org) with accession code EMD-61187. Raw data from all native MS measurements are available in Supplementary Data 1. Source data are provided with this paper.

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

## Acknowledgements

This work was supported by grants from National Natural Science Foundation of China (NSFC T2225005, 22474005 and 21927802) and National Health Commission of the People's Republic of China (2023ZD0519900, 2023ZD0520300). The authors are grateful to Prof. Junmin Quan, Yanan Lu, Siyu Liu and Yaqi Chen (Peking University Shenzhen Graduate School, China), and to Rongrong Dai (Mass Spectrometry Core, Changping Laboratory, China) for help with facility construction and technical assistance.

## Author contributions

J.W. and G.W. conceived the study. X.S., J.W., and G.W. designed and conceptualized the study. X.S., J.Y., and H.D. synthesized the materials. X.S., M.T., Y.T., H.W., C.X., Z.W., Y.Z., Y.W., L.C., and M.T. collected and analyzed the data. H.W. and Y.H. supervised the project. X.S. and M.T. wrote the manuscript. J.W. and G.W. revised the manuscript. All authors contributed to the manuscript and approved the content.

## Competing interests

J. W., Hongwei W., J. Y., and M. Tian are the inventors of a patent application based on this study (Chinese patent granted, CN201980039238.6; US and EU patent pending). The remaining authors declare no competing interests.
