## [Peer Review File · Nature Communications]

Biofunctionalized dissolvable hydrogel microbeads enable efficient characterization of native protein complexesREVIEWER COMMENTS

Reviewer #1 (Remarks to the Author):

See attachment.

Nature Communications Review

The manuscript “Biofunctionalized dissolvable hydrogel microbeads enable sensitive characterization of native protein complexes” is a well written and logic-clear manuscript. The idea applies a non-traditional elution method to wash off the target protein complex, an significant novelty of this manuscript. Before accepting the manuscript as a candidate for publication. There are still several concerns need to be addressed.

Major points:

In the passage of “Revealing individual-specific distribution of sizes and binding states of endogenous glycoprotein complexes from human sera”, it’s hard to discern the identities of health donors and patients. More info must be provided, such as the clinical significance of Hp2 pentamers, Hp2 tetramers and larger-sized oligomers.

Thermoplasma acidophilum 20S proteasome was overexpressed in E.Coli as an object to prove a successful application of the purification strategy. However, it makes no sense to spend time and words on the comparison between 20S proteasome overexpressed either in two separate plasmids but in one cell and building two genes on one plasmid. No matter how the plasmids were constructed, the T20S proteasome from E.Coli. is recombinant rather than physiological ones. This comparison doesn’t raise or answer any specific biological questions.

Author mentioned the testing the stability of recombinant complex 20S proteasome. The stability of overexpressed complex 20S could be tested by native gel and peptidase activity and Coomassie Blue Brilliant rather than applying the whole procedure of using hydrogel microbeads.

1. Figure 5A, an PsmA plasmid expression alone need to be tested as a negative control, since PsmA expression alone will generate only the double alpha-ring complex.
2. Figure 5B, authors should provide the in-gel peptidase activity to visualize the active 20S proteasome band.

Only keep the results of two genes in plasmid proteasome. That makes a typical example to prove the validity of this method.

Minor points:

1. Line #98 develop to be developed
2. Line #108, taking advantage of misspelled as taking advantages of
3. Line #131, delete one of that
4. Line #187, misspelling additionally as additionally
5. Line #235, we use a → we used a
6. Line #245, taking the advantages of should be taking advantage of
7. Line #268, of α is supposed to be of α
8. Figure 5B, the colume specification need to be provided in the main text, such as Superose 6. That will be easier for readers.

Reviewer #2 (Remarks to the Author):

The manuscript by Shao et al is a sophisticated interdisciplinary work including polymer material design and synthesis, protein purification, native mass spectrometry, and even some cryo-EM. The main claim is that specifically designed (often decorated with antibody bait) and dissolvable polymer beads (SNaP) can be used to purify the targeted proteins in order to enable native proteomics analysis of the protein complexes, often with the bait included. An interesting central argument made is that dissolving the bead matrix is more advantageous than elution or linker cleavage from conventional beads, which can allow the use of high-affinity bait-target pairs while ensuring efficient recovery of target proteins. The authors have chosen to focus more on the demonstration of this SNaP approach using several relatively simple protein systems and have not gone into more detailed native proteomics analysis. Overall, significant amount of work has been performed and I could potentially support the publication of this work in Nature Commun. if the following major and minor concerns are addressed.

I) My biggest criticism of this work is that the authors have presented insufficient data to support their major claims. The authors claimed “*Sensitive* native purification and Mass Spectrometric characterization”. However, there is no data to support the sensitivity claim (i.e. LOD, LOQ). With the fairly abundant proteins studied – e.g. Haptoglobin (Hp) is one of the most abundant proteins in the blood (around 500 mg/L or higher), also the (overexpressed) 20S proteasome is quite well-studied, how would the “sensitivity” claim be supported? I suspect that those proteins could be studied with some more common extraction/precipitation/chromatography methods without utilizing the SNaP approach. Can some protein systems that are less abundant/more challenging be targeted instead? But I am happy to leave the judgement on how useful this currently demonstrated method is to other reviewers who are the end users of this approach (and the editor).

II) At the bare minimum, the authors need to significantly tune down the claims on the utility and application of SNaP for biological and even clinical samples in various parts of the manuscript, because no experimental results are presented to support such claims. Such sentences include (but not limited to):

In abstract: “SNaP-MS shows great versatility in characterizing native complexes from complex biological samples, including clinical ones.”

In conclusion paragraph: “allowing high-affinity purification of not only overexpressed proteins from cell lysates, but also natural proteins from clinical samples.”, “serve as a general strategy for characterizing natural protein complexes in biological and clinical studies.”

III) Since the central materials innovation in the design of SNaP beads is that both crosslinkers between the linear polymers comprising the hydrogel beads and linkers connecting the baits to the beads are cleavable upon either chemical or light treatment, the presentation of Figure 1 (which is

probably the key design concept figure) can be significantly improved. I do not think I understood too much of the design ideas by looking at Figure 1. And so many references were made to many SI Schemes and Figures when the 3 different cleavage methods were discussed, it was very confusing to follow the related discussion in page 5,6.

IV) In general, the description of methods in the SI is quite simple, the authors need to add more details. One general issue is the brand and the catalog number of the biology reagent used, which would be important for others to reproduce the results. Several detailed comments below reflect this concern.

More detailed comments on various parts:

1. In Figure 1, the author claimed that the microbeads will dissolve in the elution buffer because of the cleavable linker, so what is the content of the elution buffer and what is the reaction procedure? Since the cleavage of crosslinkers and dissolution of the beads are central to the success of this approach, can the crosslinker cleavage reaction dynamic be tracked somehow. How are they sure the reaction is fully complete? If total cleavage does not happen, will this change the mass of the target in an unpredictable way or impact the downstream MS analysis?

2. In Figure 2, the author has synthesized different kinds of microbeads with different modifications, what is the zeta potential of the beads? Do they carry different charges? The authors need to have more discussion about why they have different non-specific binding capability in Figure 2A. Do they have a gel demonstrating Figure 2A? How are they detecting non-specific adsorption of proteins that are not inherently fluorescent? What proteins are they comparing against?

3. In Figure 4B, what is the SDS-PAGE gels look like of the sample before pulled down, which means what is the input sample looks like on the SDS-PAGE? The experiment method details used to stain the gel should be described.

4. In Figure 5B, how do you get the data of native PAGE? What is the buffer? What is the experiment protocol?

5. For the NMR SI figures (4,5 and 8) the peaks should be assigned to hydrogens on the molecule they are trying to prove. There is no characterization done whatsoever and the current figures look like screenshots from the instrument software). Can MS on the linkers synthesized be performed to confirm mass of these molecules?

6. In this work, much of the comparison of purification performance was made against other commercial polymer beads. It appears that the authors are not aware that nanomaterials have been developed to purify/capture/enrich (low abundance) protein samples over the last few years, for examples: Nature Commun. 11, 3903 (2020); Nature Commun. 11, 3662 (2020); Journal of the American Chemical Society, 2015, 137, 2432. I understand it is not possible to make specific performance comparison to those approaches, but these advances on protein purification should be discussed in the manuscript to provide the context for the work.

7. Figure S2. How much of each bead type are being used for enrichment? How much lysate is being used? How much protein/volume was loaded on the gel? TYPE 3 beads look like they do enrich GFP with less background proteins if they were to load more protein... Can the authors rule out that that the reason for the commercial beads to show “lower performance” were not due to poor optimization of the modification processes you performed?

8. Related to the point above, in Figure S9, why does M280 have no recovery? The authors claim there is no recovery due to irreversible binding. Irreversible binding of what to what? Biotin-Streptavidin is not irreversible interaction?

9. For Figure S6, can the background fluorescence be quantified before and after dissolving? The scale bar is not legible or not present.

Reviewer #3 (Remarks to the Author):

The paper by Shao et al describes the development of functionalized hydrogel microbeads for purification and analysis of proteins from various sources by native MS. The idea behind this approach is to overcome the need for protein elution and avoiding sample loss due to dead volume and absorbance to beads.

The manuscript falls short of meeting the publication criteria of Nature Communications due to its highly technical nature, numerous flaws, and unaddressed concerns.

Many of the experiments are ambiguous, either because it is not clear how the experiments were conducted (no appropriate details in the methods section), or insufficient controls and validations. Furthermore, the quality of most spectra provided is low, with unresolved peaks, thereby failing to support the validity and significance of the reported results.

The figures and figure legends suffer from a lack of clarity and conciseness. The legends are overly concise, failing to adequately explain the content of the figures. The figures themselves are densely packed, making it difficult to interpret the information presented. The spectra appear crowded and often overlap, obscuring the differentiation between different species. Additionally, the positioning of cartoons and labels on top of the spectra adds to the confusion, making it challenging to comprehend the experimental findings.

The manuscript lacks crucial information regarding the efficiency of the method and the impact of matrix presence on data acquisition. For instance, Figure 2B exhibits an unresolved region between 1000-4000 m/z, but the manuscript fails to address the implications of this unresolved data. Furthermore, the manuscript neglects to discuss the limitations associated with the bait being bound to the target protein and the potential effects of using high collision energy to remove it. These aspects significantly restrict the ability to further characterize the proteins of interest, particularly those susceptible to conformational changes or loss of associations and cofactor bindings at higher energies.

An example for inconsistency of the data can be seen in, Figure 5, where panel B demonstrates the presence of an intact proteasome generated from two separate plasmids, confirmed by native page, size exclusion chromatography, and electron microscopy. However, this intact proteasome is not detected by native mass spectrometry (panel C). This discrepancy between the detection of intact proteasome by alternative methods and its absence in native MS necessitates clarification and a convincing explanation to reconcile this disparity. Moreover, the absence of MS/MS analysis on the rearranged species further raises concerns. Validation is required to ensure that the “rearranged” a-subunit complex, is not actually a half proteasome containing the b-subunit pro-peptide.

Another issue of concern is the lack of novelty of the data assignment technique (Figure S14), which has been widely employed in the field for several years (PMID: 18314965). Consequently, this aspect fails to make a significant contribution to the advancement of the field.

Several inaccuracies in the references cited throughout the manuscript were identified, which diminish the overall credibility of the work and require correction.

Additionally, the manuscript lacks complete data on the theoretical and measured masses, including associated errors, for all species depicted in the figures. This crucial information is necessary to evaluate the accuracy and reliability of the reported results.

Reviewer #4 (Remarks to the Author):

The manuscript describes an exciting technology for protein purification in the native state. The work addresses a fundamental problem in protein purification/pulldown workflow, tags that are extremely high-affinity and provides high-fidelity in substrate capture are often irreversible. This leads to difficulty in eluting the substrate from the bait post-pulldown, especially under native conditions. The authors here claim to solve this problem by developing universally applicable chemically/photochemically dissolvable microbeads, termed SNaP. I have several major concerns/questions about the claims the authors have made. However, if these concerns are addressed satisfactorily in the revised version, I would strongly support the publication of this manuscript. Below are my specific concerns.

General comment:

1. In some places, the authors veer off in discussing aspects that are not directly related to the question at hand. For example, in the second paragraph page 2, on glycosylation. Detecting well-resolved MS from heterogeneous PTM samples is a major technical challenge in native MS. But that does not have anything to do with purification. An epitope that recognizes all proteoforms will carry the same problem, irrespective of how efficient the enrichment strategy is. A good example is the authors' case study with the Hp-Hb system. An efficient enrichment did not solve the mass heterogeneity problem. So, discussing this in the middle of the intro breaks the flow.

2. The overall strategy of synthesizing these beads seems very straightforward. If the authors' scientific claims are right, then this would certainly be a major step forward from the standard agarose beads used. So for the widespread use of the Authors' protocol, I recommend both the synthesis, as well as microbead preparation sections be further elaborated with detailed preparation protocols, troubleshoot, and checkpoints. That would also certainly broaden the applicability of the current work/strategy.

GFP-capturing experiment:

1. Here the authors used thiol-linked GFP-antibody for pulldown and subsequent reduction for elution. The manuscript does not describe how the antibody was modified. The SI section only details the DBCO modification, not the modification by the thiol-linked linker. Since the authors are using this as a benchmark to showcase the advantages of their approach, details of this, including prior references from which this strategy was adopted must be included.

2. It is also a bit surprising that the authors used GFP-antibody here as the gold standard to pull down GFP. In practice, this is not right. The most used bait to pull down GFP is the variable domain GFP-nanobody. Is there a reason why that was not used?

Synthesis of dissolvable beads:

1. Overall the synthesis section is difficult to follow. A general comment is to write the name of the chemicals used in the steps so that non-specialist users can follow the chemistry. A good example of that is Figure S3. Please provide the name of the chemicals on top of the arrow. Also refer to these figures, while describing the synthesis process in the earlier SI-section (between pages S2-S7).

2. While adequate NMR spectra have been shown to confirm the synthesis of the bead material, there is little quality control shown on the microbeads themselves. At least some TEM images of the beads and comparison with the commercial beads would be necessary.

3. Figure S3 legends need to be expanded. It is a key figure.

4. The fabrication step of the microbeads and any related quality control should be elaborated

Strep-GFP-capturing experiment from TAMARA labeled lysate:

1. Line 151/152 should cite Figure S3 for the GFP-strep modification protocol

2. Again, the authors need to justify why they are using GFP-antibody, instead of the more commonly used GFP-nanobody; especially when protocol is available to couple nanobody to beads using thiol or other chemistry (PMID: 26633879)

3. Figure 2A follows what is expected based on the binding affinities. But that has nothing to do with the developed SNAP beads. The real power of the technique is that it can use the highest affinity binding to capture the target with the least background, which other agarose beads also can, but then can subsequently elute it in native conditions, which one cannot do from standard agarose functionalized beads. So the "money-shot" here is not the comparison between different tags with the same SNAP beads but between the highest affinity elutable tag in agarose beads vs. the highest affinity tag (otherwise non-elutable) in SNAP beads. Among the tags used here, streptavidin is the

highest affinity tag that allows native elution using desbiotin from standard agarose beads. Streptavidin, while offering the highest affinity, cannot be eluted from agarose. So, the authors should compare the capture/background ratio between Streptavidin-SNaP and Streptactin-Agarose. Part of that data is in Figure S9. While that could stay there, Figure 2 should include a similar fl.-based GFP/background capture ratio for Streptactin-agarose.

4. Following the same notion, the authors should compare side-chain cleavable solid agarose beads having the Streptavidin bait.

5. Both here, as well as in subsequent sections, the authors have used native MS to demonstrate the fidelity of the native elution process (by showing non-covalent homomeric and heteromeric complexes), as well as the purity of the eluted. While native MS can certainly confirm the first, it is not the most suitable approach for demonstrating the second. This is because in native MS it is feasible to tune MS in a manner that makes it more sensitive towards a particular m/z region. In UHMR, the instrument used here, change in trapping gas Pressure and/or rf voltages can significantly bias the spectra towards high/low mass region. So for assessing the quality of the purified samples, conventional bottom-up proteomics to see how target proteins are enriched over the background can still provide the most in-depth view of the purification process. So to fully convince readers to switch from their established agarose bead approach to SNaP beads a simple LFQ-based bottom-up proteomics approach will go a long way. The authors can take Streptavidin-SNaP eluted samples and Streptactin-Agarose eluted samples. Using an LFQ-based study can show the degree of enrichment of GFP over the background in both cases. This will also broaden the use of the method, as there are certainly more labs doing pull-down proteomics to find interactors than doing nativeMS. A very high-affinity elutable tag would certainly benefit that endeavor too.

Compatibility of recovered protein sample with native MS characterization

1. This is an important point, as it seems eventually for native MS one has to do another round of purification. Is the SEC a must? Zeba/bio spin-based spin columns are typically used for quick buffer exchange from Tris/Phosphate buffers to native MS-compatible AmAc. Can they not remove the polymer? One exciting prospect of this high-affinity/high-efficiency capture of SNaP beads is that one can purify native MS-worthy material from a very small amount of starting samples. However, having SEC as a necessary step in this pipeline makes it slightly more sample-demanding. It would be nice if authors could circumvent this need

2. Also, SEC itself is a protein purification process. So if it is a must step, then the comparison shown in Figure 2 has to be done at the end SEC of the respective samples.

Avidin-capturing experiment

1. For r-Avidin (figure 3B), the monomer seems to be emerging at a higher in-source. Is that because background polymer requires more in source for better desolvation?

2. Overall, the peaks look rather broad, especially for tetramer. The authors said that they used in-source energy. UHMR also has a front-end trapping-based activation, termed desolvation energy,

which is more effective than standard insource. Have authors tried using it to see if they could get better-resolved spectra?

3. Due to possible effects on the intramolecular disulfide bonds, I thought this was the perfect example to use the UV-dissolvable beads. The authors mentioned making it in the first section but never used it. If they want to claim that they have also made UV-dissolvable SNaP beads, they need to use them here.

Hp-Hb capturing experiment

1. In Figure 4B, the charge state assignment should be made to the MS. In P3, the peaks are not resolved enough to annotate the charge state assignment.

2. Here, the complexity, arises from glycosylation. Since the SNaP tag allows native-complex binding, would it not be possible to do a de-glycosylation, on the bead by enzyme such as PNGase? Post-reaction, the beads can be separated from the free glycan and PNGase by centrifuging. Subsequently dissolved and subjected to MS.

DETAILED POINT BY POINT RESPONSE

Reviewer 1

The manuscript “Biofunctionalized dissolvable hydrogel microbeads enable sensitive characterization of native protein complexes” is a well written and logic-clear manuscript. The idea applies a non-traditional elution method to wash off the target protein complex, an significant novelty of this manuscript. Before accepting the manuscript as a candidate for publication. There are still several concerns need to be addressed.

- We are grateful for the reviewer’s constructive comments, which have greatly improved the quality of this manuscript.

Major points:

In the passage of “Revealing individual-specific distribution of sizes and binding states of endogenous glycoprotein complexes from human sera”, it’s hard to discern the identities of health donors and patients. More info must be provided, such as the clinical significance of Hp2 pentamers, Hp2 tetramers and larger-sized oligomers.

- We agree with the reviewer’s suggestion to provide a more detailed description of the individuals who contributed blood samples for the Hp distribution profiling. The experiments were not designed as a cohort study, and we did not intentionally include any previously defined patients among the blood donors. These donors, who are also authors of this manuscript, were not identified as having anhaptooglobinemia until we observed a low abundance of Hp in the blood of Donor P4. We have revised the manuscript to include the following description in the Methods section: “Blood samples were collected from four donors, none of whom had been previously diagnosed with any health conditions relevant to this study. All donors provided written informed consent prior to participation.” We have also added the following description in the Results section: “In this study, we collected blood samples from four donors (P1-P4) who had not been diagnosed with any health conditions relevant to this study.”
- We also agree with the reviewer’s suggestion to provide more information on the clinical significance of the larger oligomer of Hp. We have added the following description in the Discussion section: “In addition to its critical role in binding free Hb for tissue protection and oxidative damage prevention, Hp is implicated in the prevalence and clinical progression of numerous inflammatory diseases, including infections, atherosclerosis, and autoimmune disorders¹. Prior studies have suggested a potential association between Hp polymorphism and diseases such as diabetes and albuminuria^{2, 3, 4}. Furthermore, Hp is a potential biomarker for diseases including various forms of malignant neoplasms⁵.”

Thermoplasma acidophilum 20S proteasome was overexpressed in E.Coli as an object to prove a successful application of the purification strategy. However, it makes no sense to spend time and words on the comparison between 20S proteasome overexpressed either in two separate plasmids but in one cell and building two genes on one plasmid. No matter how the plasmids were constructed, the T20S proteasome from E.Coli. is recombinant rather than physiological ones. This comparison doesn’t raise or answer any specific biological questions.

- We appreciate the suggestion to clarify our experiment descriptions and discussions. In response, we have revised the relevant sections. One of the designed applications of SNaP-MS is swiftly screening optimal expression and sample preparation strategies for overexpressed protein systems, with the goal of facilitating their characterization in structural biology. We utilized 20S proteasome, a model commonly used in the structural biology community, to demonstrate that SNaP-MS allows for concurrent characterization of the integrity and degradation products, including rearranged subcomplexes and the dissociation products. This is achieved using native MS, which is more accessible than conventional structural biology tools like cryo-EM, considering cost and operation requirements. The 20S complexes, expressed from a single plasmid and two separate plasmids respectively, showed significant differences in resistance to degradation during sample preparation. Although these 20S complex formats did not directly address a specific biological question within the context of method development, they served as models that demonstrated the technical value of SNaP-MS in enhancing the quality of data to be acquired by structural biology tools and improving their workflows. This, in turn, aids in addressing more specific biological questions using these tools. We have revised the corresponding paragraphs in both the Results and Discussion sections for improved clarity.

Author mentioned the testing the stability of recombinant complex 20S proteasome. The stability of overexpressed complex 20S could be tested by native gel and peptidase activity and Coomassie Blue Brilliant rather than applying the whole procedure of using hydrogel microbeads.

- We are grateful to the reviewer for suggesting the stability tests using gel electrophoresis-based approaches. In response to this suggestion, we conducted a native gel analysis of Avi-, His-, and Strep-tagged 20S samples to assess their stability. The results, which corroborated the findings from SEC and EM, have been incorporated into the revised Figure 5B. Furthermore, we carried out an in-gel peptidase activity assay of 20S purified by SNaP beads. This was compared with the untreated cell lysate containing 20S, the dissolving products of SNaP beads conjugated with SA, and 20S captured by conventional solid beads. The results, which indicate comparable activity of 20S purified by SNaP beads and conventional solid beads, have been included in the new Figure S24 in the revised SI.

1. Figure 5A, an PsmA plasmid expression alone need to be tested as a negative control, since PsmA expression alone will generate only the double alpha-ring complex.

- In line with the reviewer's suggestion, we utilized the same vector to construct the PsmA plasmid that only codes for 20S α subunit, and used the self-assembled products of the expressed α subunits as a negative control for the complex integrity analysis. We detected the heptamer and 14-mer of the α subunit, which consist of a double-layer of α -rings, in native MS. Their identities were verified by tandem MS. These data are presented in the newly added Figure S23 in the revised SI.

2. Figure 5B, authors should provide the in-gel peptidase activity to visualize the active 20S proteasome band.

Only keep the results of two genes in plasmid proteasome. That makes a typical example to prove the validity of this method.

- In accordance with the reviewer's suggestion, we conducted an in-gel peptidase activity assay of 20S purified by SNaP beads. This was compared with the untreated cell lysate containing 20S, the dissolving products of SNaP beads conjugated with SA, and 20S captured by conventional solid beads. The compositions of these samples were confirmed with SDS-PAGE. Additionally, we performed an in-solution activity assay of 20S purified by SNaP beads, with the dissolving products of SA-conjugated SNaP beads serving as the negative control. The results, which indicate comparable activity of 20S purified by SNaP beads and conventional solid beads, have been incorporated into the new **Figure S24** in the revised SI.

Minor points:

- We are grateful to the reviewer for his thorough reading. We have meticulously corrected the typographical errors and minor mistakes in the previous version, including the following ones identified by the reviewer.

1. Line #98 develop to be developed

- Corrected.

2. Line #108, taking advantage of misspelled as taking advantages of

- Corrected.

3. Line #131, delete one of that

- Corrected.

4. Line #187, misspelling additionally as additionally

- Corrected.

5. Line #235, we use a we used a

- Corrected.

6. Line #245, taking the advantages of should be taking advantage of

- Corrected.

7. Line #268, of α is supposed to be of α

- Corrected.

8. Figure 5B, the column specification need to be provided in the main text, such as Superose 6. That will be easier for readers.

- We have included the specification of both SEC and WCS columns used in this study in the paragraph describing "Liquid chromatography (LC)" in the Methods section, which has now been relocated from SI to the main text.

Reviewer 2

The manuscript by Shao et al is a sophisticated interdisciplinary work including polymer material design and synthesis, protein purification, native mass spectrometry, and even some cryo-EM. The main claim is that specifically designed (often decorated with antibody bait) and dissolvable polymer beads (SNaP) can be used to purify the targeted proteins in order to enable native proteomics analysis of the protein complexes, often with the bait included. An interesting central argument made is that dissolving the bead matrix is more advantageous than elution or linker cleavage from conventional beads, which can allow the use of high-affinity bait-target pairs while ensuring efficient recovery of target proteins. The authors have chosen to focus more on the demonstration of this SNaP approach using several relatively simple protein systems and have not gone into more detailed native proteomics analysis. Overall, significant amount of work has been performed and I could potentially support the publication of this work in Nature Commun. if the following major and minor concerns are addressed.

- We are very thankful to the reviewer for the favorable comments. In line with the reviewer's suggestion, we have conducted proteomic analysis to support the corresponding conclusions. Please refer to our responses to the corresponding comments for more details.

*1) My biggest criticism of this work is that the authors have presented insufficient data to support their majors claims. The authors claimed “*Sensitive* native purification and Mass Spectrometric characterization”. However, there is no data to support the sensitivity claim (i.e. LOD, LOQ). With the fairly abundant proteins studied – e.g. Haptoglobin (Hp) is one of the most abundant proteins in the blood (around 500 mg/L or higher), also the (overexpressed) 20S proteasome is quite well-studied, how would the “sensitivity” claim be supported? I suspect that those proteins could be studied with some more common extraction/precipitation/chromatography methods without utilizing the SNaP approach. Can some protein systems that are less abundant/more challenging be targeted instead? But I am happy to leave the judgement on how useful this currently demonstrated method is to other reviewers who are the end users of this approach (and the editor).*

- We agree with the reviewer that a more detailed and precise explanation to the claims regarding sensitivity is necessary to prevent confusion, and additional data are needed to support the sensitivity claim.
- In this work, the term ‘sensitivity’ refers to the ability to effectively purify target protein complexes present in small amounts and at low concentrations in a low volume and quantity of biological samples for effective downstream biophysical characterization.
 - Concerning the input quantity of biological samples: SNaP-MS enables effective purification of target protein complexes from a mere 100 μ L of cell lysate or human serum (for all samples measured in this study; described in the Methods section under the subheading of “Purification and recovery of target protein”).
 - With respect to the initial concentration of the target: SNaP-MS facilitates effective purification of model proteins (such as GFP) at a concentration as low as 10 ng/ μ L in cell lysate (Figure 2D).

- Pertaining to the initial quantity of the target: SNaP-MS allows effective purification of 20S at the picomole level in the cell lysate (Figure S22).
- Although Hp is among the abundant proteins in plasma, it exhibits high heterogeneity in terms of polypeptide chain lengths, glycosylation, oligomeric states, and binding states, unlike relatively homogeneous proteins such as albumin. The co-existence of these modification and processing states divides a seemingly single protein species into a collection of proteoforms with different masses, which greatly dilutes the detectable signal of each proteoform. In the case of Hp, whose heavy chain has four N-linked glycosylation sites and more than 100 glycans have been identified at each site, the combination of these glycans, along with a range of Hp oligomeric states and binding states, actually reduces the effective concentration of each proteoform by several orders of magnitude. Therefore, at the intact protein level, Hp should be regarded as a collection of less abundant proteoforms, rather than a single abundant protein. This is in contrast with most existing studies where Hp was characterized at the peptide or subunit level. Therefore, Hp serves as a good model to demonstrate the capability of SNaP-MS in capturing highly heterogeneously distributed less abundant proteoforms in their native forms from blood. We have added this discussion in the Results section.
- Although 20S was overexpressed, the low input quantity (picomole level) of 20S in a low input volume (100 μ L) of cell lysate required for effective purification and characterization can still demonstrate the sensitivity of SNaP-MS. This low input circumvents the conventional culture scale and allows utilization of petri-dish for rapid and convenient protein expression. This system serves as a model to demonstrate the application of SNaP-MS in characterizing the overexpressed proteins from cell lysate, which are frequently employed in structural biology and biophysics. We have added this discussion to the revised manuscript.
- To provide an estimation of the “LOD” of SNaP-MS, we conducted protein quantification and sequential dilution of the Hp sample purified from Individual P1. As shown in the new Figure S20 in the revised SI, the critical dilution fold that results in adequate spectral quality for reliable data interpretation lies between 2x and 3x. Since the protein concentration of the undiluted sample that underwent SNaP purification was determined to be 5.8 μ g/mL with a BCA assay, we suggest that proteins with concentrations higher than 3 μ g/mL could be effectively detected by native MS. Considering that Hp is a highly heterogeneous protein system (as discussed above), the minimal concentration of a relatively homogeneous protein needed for effective characterization can be significantly lower. We have added this discussion in the Results section.

II) At the bare minimum, the authors need to significantly tune down the claims on the utility and application of SNaP for biological and even clinical samples in various parts of the manuscript, because no experimental results are presented to support such claims. Such sentences include (but not limited to):

In abstract: “SNaP-MS shows great versatility in characterizing native complexes from complex biological samples, including clinical ones.”

In conclusion paragraph: “allowing high-affinity purification of not only overexpressed proteins from cell lysates, but also natural proteins from clinical samples.”, “serve as a general strategy for characterizing natural protein complexes in biological and clinical studies.”

- We concur with the reviewer that the term “clinical sample” should be carefully reconsidered and rephrased to avoid any misunderstanding. We initially used this term because we purified and characterized Hp with SNaP-MS from actual blood samples of multiple individuals, and obtained clinically relevant information, including the individual-specific oligomeric state distribution and anhaptoalbuminemia of Individual P4, who had not been diagnosed prior to the SNaP-MS measurements. However, we acknowledge that these blood samples were not collected as part of a systematically designed clinical study, and whether these blood samples can be classified as clinical ones may be subject to debate. As suggested by the reviewer, we have rephrased the corresponding sentences, including those highlighted by the reviewer, by substituting “clinical sample” with “biological sample” or “blood sample”. We have also discussed their potential application to clinical samples in the Discussion section of the revised manuscript

III) Since the central materials innovation in the design of SNaP beads is that both crosslinkers between the linear polymers comprising the hydrogel beads and linkers connecting the baits to the beads are cleavable upon either chemical or light treatment, the presentation of Figure 1 (which is probably the key design concept figure) can be significantly improved. I do not think I understood too much of the design ideas by looking at Figure 1. And so many references were made to many SI Schemes and Figures when the 3 different cleavage methods were discussed, it was very confusing to follow the related discussion in page 5,6.

- We express our gratitude to the reviewer for this insightful observation. We have revised the entire **Figure 1** by reorganizing the materials and incorporating annotations for a clearer presentation of the design of the SNaP beads.

IV) In general, the description of methods in the SI is quite simple, the authors need to add more details. One general issue is the brand and the catalog number of the biology reagent used, which would be important for others to reproduce the results. Several detailed comments below reflect this concern.

- We agree with the reviewer that more experimental details are necessary for the Method section. We have not only incorporated the suggested details but also added paragraphs describing the additional experimental details in the Methods section, which has been relocated from the original SI to the main text in the revised manuscript.

More detailed comments on various parts:

1. In Figure 1, the author claimed that the microbeads will dissolve in the elution buffer because of the cleavable linker, so what is the content of the elution buffer and what is the reaction procedure? Since the cleavage of crosslinkers and dissolution of the beads are central to the success of this approach, can the crosslinker cleavage reaction dynamic be tracked somehow. How are they sure the reaction is fully complete? If total cleavage does not happen, will this change the mass of the target in an unpredictable way or impact the downstream MS analysis?

- We concur with the reviewer that the details of bead dissolving should be provided. We have included such details in the revised Methods section under the subheading of “Purification and Recovery of Target Protein”. The SNaP beads were dissolved at room temperature either by incubating with 5 mM TCEP (final concentration) and rotating for 10 min (for the chemically

dissolvable beads), or by exposure to 365 nm UV light for 30 sec (for the photo-dissolvable beads).

- There were two types of cleavable linkers used in the SNaP beads, *i.e.* the bead-bait linker and the inter-polymer crosslinker. The cleavage of the former is responsible for the release of the target proteins, while the cleavage of the latter is responsible for the elimination of bead *per se*. Since the bead-bait crosslinkers are located peripherally on the beads, they are preferentially cleaved compared with the inter-polymer crosslinkers. We employed multiple techniques, including dynamic light scattering (DLS), brightfield and fluorescence microscopy, and SDS-PAGE, to track and characterize the crosslinker cleavage reaction and the bead dissolving process. The microscopic images shown in Figure 1C and the revised Figure S9A in SI suggest the disappearance of the beads upon chemical and UV-light treatment, respectively. The DLS results shown in Figure S10 indicate the complete elimination of large particles upon the dissolving of SNaP beads with 10 mM TCEP. The SDS-PAGE results shown in the revised Figure S9B indicate that neither the baits nor the purified targets underwent artificial association after bead dissolution, and no proteins were detected in conjugation with the hydrogel polymer. These results demonstrate the completion of the cleavage reaction required for target release under the experimental conditions employed in this work. We have added the corresponding descriptions in both the Results and Discussion sections of the revised manuscript.

2. In Figure 2, the author has synthesized different kinds of microbeads with different modifications, what is the zeta potential of the beads? Do they carry different charges? The authors need to have more discussion about why they have different non-specific binding capability in Figure 2A. Do they have a gel demonstrating Figure 2A? How are they detecting non-specific adsorption of proteins that are not inherently fluorescent? What proteins are they comparing against?

- We express our gratitude to the reviewer for suggesting the measurement of the zeta potential of beads. We have conducted such measurements and presented the results in the new Figure S8 in SI. The data indicate that the surface of beads with exposed amino or and azide group carry a positive charge, while that of sdAb-conjugated beads carry a negative charge.
- For the non-specific adsorption measurements shown in the original Figure 2A (repositioned as Figure 2D in the revised version), the background proteins were modified with the fluorescent label TAMRA-NHS for fluorescence detection. This was followed by the removal of excess fluorescent molecules through centrifugal ultrafiltration. These labeled background proteins were then mixed with the target protein to measure the non-specific adsorption. We have added these experimental details in the caption of the new Figure 2.
- In response to the reviewer's suggestion, we have performed SDS-PAGE of GFP samples purified from *E. coli* lysate using SNaP beads conjugated with different baits. The results are included in the new Figure 2 and new Figure S11 in SI. The relative abundances of the captured GFP in comparison with the background proteins could be visualized.
- To validate the specificity evaluation results, we have performed a quantitative proteomics analysis of GFP samples purified from *E. coli* lysate using SNaP beads modified with different baits. We measured the abundances of peptides from GFP to evaluate the purification

efficiency and measured the relative abundance ratios of peptides from GFP to those from the background proteins to evaluate the specificity. As shown in the new Figures 2E and 2F, the proteomic results support the results in the original Figure 2A (repositioned as Figure 2D in the revised version). The experimental details of MS-based label-free quantitative proteomics have been included in the Methods section.

- In the purification products, the detected background proteins may encompass those non-specifically bound by the bait and those adsorbed onto the bead surface. Our proteomic analysis has revealed the identities and relative quantities of these background proteins, which were captured by SNaP beads conjugated with different baits (the new Figure S26). The reproducibility of the results, derived from three measurements for each bead type, suggests that the capture of background proteins is not entirely random. The abundance distributions of these detected background proteins display bait-specific patterns, indicating that the bait identity influences the non-specific capture. The isoelectric point (PI) distributions of these background proteins imply that there are other factors, apart from electrostatic interaction, that drive the non-specific capture. Consequently, the observed correlation between the detected quantity of background proteins and the affinity of the bait conjugated on the beads (the new Figure 2) can be partially attributed to the specificity of each bait. Therefore, the enhanced flexibility in bait selection, facilitated by the SNaP-MS scheme, promotes not only the target recovery rate but also the specificity of purification. We have incorporated such discussion on the different non-specific binding capabilities in the revised Discussion section.

3. In Figure 4B, what is the SDS-PAGE gels look like of the sample before pulled down, which means what is the input sample looks like on the SDS-PAGE? The experiment method details used to stain the gel should be described.

- We concur with the reviewer that the SDS-PAGE of the input serum sample prior to pulling should be presented, in addition to the purified proteins. We have updated Figure 4B to include SDS-PAGE images of not only Hp purified from different donors' sera but also an untreated serum sample from Individual P1.
- Upon completion of the electrophoresis, protein staining was performed using the eStain[®] L1 protein staining system (Genescript) following the standard staining protocol, which includes two staining and two de-staining cycles. We have incorporated these details into the paragraph on SDS-PAGE in the Methods section of the revised manuscript.

4. In Figure 5B, how do you get the data of native PAGE? What is the buffer? What is the experiment protocol?

- We have included the experimental details of native PAGE analysis in the Methods section of the revised manuscript.

5. For the NMR SI figures (4,5 and 8) the peaks should be assigned to hydrogens on the molecule they are trying to prove. There is no characterization done whatsoever and the current figures look like screenshots from the instrument software). Can MS on the linkers synthesized be performed to confirm mass of these molecules?

- In line with the reviewer's suggestion, we have updated all the NMR spectra (now presented in the new **Figures S2-S4** in SI) by labeling the peaks with hydrogen assignments. The locations of hydrogens assigned to the labeled peaks are indicated in the structural formula shown as insets. Additionally, we have replaced all the original figures generated by the instrument software with high-quality vector versions
- We appreciate the reviewer's suggestion regarding the MS characterization of the linkers. We have performed both intact mass (MS1) and tandem mass (MS2 and MS3) measurements of these linkers. The masses of the intact molecules and the fragmentation behavior of both the intact molecules and their predominant fragments have verified the structures of these molecular species. These additional mass spectra have been included in the new **Figures S2 and S4** in SI.

6. In this work, much of the comparison of purification performance was made against other commercial polymer beads. It appears that the authors are not aware that nanomaterials have been developed to purify/capture/enrich (low abundance) protein samples over the last few years, for examples: Nature Commun. 11, 3903 (2020); Nature Commun. 11, 3662 (2020); Journal of the American Chemical Society, 2015, 137, 2432. I understand it is not possible to make specific performance comparison to those approaches, but these advances on protein purification should be discussed in the manuscript to provide the context for the work.

- We express our gratitude to the reviewer for the insightful comments on nanomaterials. The approaches to isolate or enrich low-abundant proteins in biological samples using nanomaterials are not compatible with native conditions, and therefore, are not suitable for the application scenarios of SNaP-MS. We have incorporated the corresponding discussion and references about nanomaterials for protein purification into the Discussion section of the revised manuscript.

7. Figure S2. How much of each bead type are being used for enrichment? How much lysate is being used? How much protein/volume was loaded on the gel? TYPE 3 beads look like they do enrich GFP with less background proteins if they were to load more protein... Can the authors rule out that that the reason for the commercial beads to show "lower performance" were not due to poor optimization of the modification processes you performed?

- We express our gratitude to the reviewer for this insightful comment. We have incorporated these experimental details into the Methods section of the revised manuscript.
- To ensure a fair comparison, we employed the same click-chemical reaction (azide-DBCO) to modify different carriers with the bait protein and conducted purification with the same input quantity of samples under meticulously optimized conditions. Consequently, the efficiency of the modification and cleavage reactions on the surface of different carriers should be the primary factor influencing the bead modification and purification. In a comparison of the purification efficiencies provided by SNaP beads and side-chain cleavable agarose beads, where the same bait and modification reactions were utilized, a significant difference in purification recovery was observed (as shown in the new **Figure 2B** and **Figure S21** in SI), which supports our conclusion.

8. Related to the point above, in Figure S9, why does M280 have no recovery? The authors claim there is no recovery due to irreversible binding. Irreversible binding of what to what? Biotin-Streptavidin is not irreversible interaction?

- We appreciate the reviewer's suggestion to clarify our phrasing. The biotin-streptavidin interaction is among the strongest non-covalent interactions known to date. Although it can be chemically reversed, the reversal requires harsh conditions that also denature the target protein. To release the target proteins under non-denaturing conditions, conventional non-denaturing affinity purification requires competitive elution, which involves replacing the original bait-target interaction with a higher affinity binding with the bait. However, the high-affinity binding between streptavidin and biotin can hardly be replaced under non-denaturing conditions. This is one of the reasons we developed the SNaP beads, which circumvent this problem by substituting the conventional competitive elution with bait release. For better clarification, we have rephrased the corresponding discussion by replacing "irreversible binding" with "the high-affinity binding between SA and biotin, which prevented the release of the biotin-modified GFP through competitive elution under non-denaturing conditions".
- The original Figure S9 has been revised and repositioned as Figure 2A in the revised manuscript.
- To validate the recovery rate results measured based on fluorescence, we performed additional quantitative proteomics analysis. We measured the ratio of the average abundance of peptides from GFP to those from the background proteins, where GFP was purified with SNaP beads, Strep-Tactin XT agarose beads (ST-XT, IBA Lifesciences), and M280 streptavidin modified magnetic beads (M280, Thermo Fisher Scientific) respectively from *E. coli* lysate (new Figure 2B). The data of purification with M280 beads verified the low recovery measured based on fluorescence (new Figure 2A)

9. For Figure S6, can the background fluorescence be quantified before and after dissolving? The scale bar is not legible or not present.

- Fluorescence serves as a valuable tool for visualizing the shape of the beads and assessing the extent of bead dissolving. However, accurately quantifying the background fluorescence before and after bead dissolution presents a challenge due to the low intensity of background signals. Acquisition of such signals requires high exposure of excitation light, which increases the photobleaching of the fluorophores and leads to unreliable quantitation results. As an alternative approach, we utilized Dynamic Light Scattering (DLS) data, as shown in the new Figure S10 in SI, to demonstrate the conversion of hydrogel beads into dissolvable particles.
- We have revised the original Figure S6, which has been repositioned as Figure S9A, by updating the scale bars.

Reviewer 3

The paper by Shao et al describes the development of functionalized hydrogel microbeads for purification and analysis of proteins from various sources by native MS. The idea behind this approach is to overcome the need for protein elution and avoiding sample loss due to dead volume and absorbance to beads.

The manuscript falls short of meeting the publication criteria of Nature Communications due to its highly technical nature, numerous flaws, and unaddressed concerns.

- We express our gratitude to the reviewer for the constructive feedback, which has significantly contributed to enhancing the quality of our manuscript. It appears from Reviewer 3's comments that there may be several misunderstandings. We hope that the improved presentation in our revised manuscript will help to clarify these points.
- In line with the Aims & Scope of Nature Communications, which states that "Papers published by the journal aim to represent important advances of significance to specialists within each field", we believe that the "highly technical nature" of our work does not exclude it from the scope of Nature Communications. In response to the flaws and concerns raised by the reviewers, we have conducted additional experiments and made significant revisions to address these issues. For more details, please refer to our point-by-point responses to the corresponding comments.

Many of the experiments are ambiguous, either because it is not clear how the experiments were conducted (no appropriate details in the methods section), or insufficient controls and validations. Furthermore, the quality of most spectra provided is low, with unresolved peaks, thereby failing to support the validity and significance of the reported results.

- We appreciate the reviewer's comments on the clarity of our experimental descriptions. In response, we have thoroughly revised the Methods section, which has been moved from the original SI to the main text. This revision includes necessary experimental details and rephrased descriptions for enhanced clarity.
- Regarding controls and validations, we have conducted additional experiments to provide the necessary controls and validations. These include experiments carried out with gel electrophoresis, in-gel activity assay, organic mass spectrometry, native mass spectrometry, quantitative proteomics, and fluorescence microscopy. Please refer to our point-by-point responses to the corresponding comments for more details.
- The reviewer's comment on the "low quality" of spectra with unresolved peaks may stem from a misunderstanding. This has made us realize that we need to improve the clarity of our presentation to prevent readers from confusing the mass spectra of intact heterogeneous glycoproteins with those of homogeneous non-glycosylated proteins or those of portions of glycoproteins.

Some proteins characterized in this work, such as haptoglobin (Hp) and avidin, are glycoproteins that exhibit significant mass heterogeneity. The diversity in the carbohydrate residue masses and the glycan chains that differ in the compositions of these residues result in an intrinsically broad distribution of the masses of the intact glycoproteins and broadened

signal peaks. Since the intact protein ions are multiply charged in ESI and exhibit multiple charge states, when the distribution width of the m/z values of protein ions in a specific charge state exceeds the spacing between the centroids of signals of adjacent charge states, these signals are intrinsically convoluted, regardless of the resolving power of the mass spectrometers, experimental methods, or data acquisition parameters. In the case of Hp, the spectra can be further complicated by the high heterogeneity in terms of chain lengths, oligomerization states, and binding states, in addition to glycosylation. This can be illustrated by a number of published spectra such as Figure 1C of *Anal. Chem.* 2017, 89, 4793–4797 and Figure 6(a) and 6(b) of *Cell. Mol. Immunol.* 2019, 16, 460–472. This is in sharp contrast with the non-glycosylated proteins, which exhibit defined masses and well-separated signal peaks.

Considering this high heterogeneity and compared with the previously published data of intact glycoproteins, the mass spectra presented in this work are actually of high quality. As introduced and discussed in the Introduction section, approaches such as limited charge reduction, heterogeneity dilution, and charge detection MS (CDMS) were developed to solve this problem. However, these approaches require either specific models of mass spectrometers to provide gas-phase reaction functions for charge manipulation or antibodies that specifically bind the analyte glycoproteins and undergo specific fragmentation during collision-based dissociation, which greatly limit the application of these methods.

In contrast, the workflow of SNaP-MS allows attachment of the bait protein to the target glycoproteins, and the bait can serve as the charge remover and mass corrector, greatly increasing the charge resolution of the glycoprotein signals through charge reduction and increasing the accuracy of mass determination through mass balance of the dissociation reaction, *i.e.*, the bait-target mass equals the bait mass plus the target protein mass (refer to the new Figure S18 in SI and the updated Discussion section). This allows reliable mass determination and identification with a broader range of mass spectrometers. In other words, the SNaP-MS workflow provides a means to improve, rather than lowers the quality of native mass spectra of intact heterogeneously glycosylated proteins.

The spectral qualities of two formats of avidin, which are less heterogeneously glycosylated proteins, allows signals of various proteoforms to be isotopically resolved. This facilitated the profiling of proteoform distributions, as shown in Figure 3D. The spectral qualities of non-glycosylated proteins, such as 20S proteasome, as presented in Figure 5 and the new Figure S23, surpass those in previously published studies on the same system in terms of spectral resolution and depth of identification^{6, 7, 8}.

The figures and figure legends suffer from a lack of clarity and conciseness. The legends are overly concise, failing to adequately explain the content of the figures. The figures themselves are densely packed, making it difficult to interpret the information presented. The spectra appear crowded and often overlap, obscuring the differentiation between different species. Additionally, the positioning of cartoons and labels on top of the spectra adds to the confusion, making it challenging to comprehend the experimental findings.

- We express our gratitude to the reviewer for highlighting this issue. In response, we have undertaken a comprehensive revision of all figures and their corresponding captions to enhance the clarity of presentation by integrating more detailed information and reorganizing the materials within the figures.

The manuscript lacks crucial information regarding the efficiency of the method and the impact of matrix presence on data acquisition. For instance, Figure 2B exhibits an unresolved region between 1000-4000 m/z, but the manuscript fails to address the implications of this unresolved data. Furthermore, the manuscript neglects to discuss the limitations associated with the bait being bound to the target protein and the potential effects of using high collision energy to remove it. These aspects significantly restrict the ability to further characterize the proteins of interest, particularly those susceptible to conformational changes or loss of associations and cofactor bindings at higher energies.

- We express our gratitude to the reviewer for their insightful comments on the efficiency of the method, the impact of the bead matrix on data acquisition, and the influence of bait attachment.
- Regarding efficiency, we have supplemented the data shown in the original Figures 2, S2, S6, S7, and S9 with additional experimental data in the new Figures 2B, 2E, 2F, S21, and S24. These were obtained using gel electrophoresis, in-gel activity assay, and quantitative proteomics. These data demonstrate the higher efficiency of the method.
- In terms of the matrix's impact, we have expanded upon the original text (Lines 164-169) and data (Figures S10-S12) with additional descriptions and discussions on matrix removal and the effect of matrix dissolving products on target protein characterization. Regarding reviewer's mentioning of "Figure 2B", we believe the reviewer intended to refer to Figure 3B, instead of Figure 2B, as the 1000-4000 m/z region was not included in the spectra shown in Figure 2B. The signals in this region are caused by linear polymers, the dissolving product of SNaP beads. Although detectable at the native MS stage, these signals did not interfere with the characterization of intact large protein complexes due to non-overlapping m/z distribution regions. These signals also did not interfere with the characterization of released protein subunits, as these polymers either do not survive the in-source fragmentation energies that are sufficient to dissociate the complexes in the quasi-MS2 mode (without mass-selection of the precursor ions), or can be filtered out during mass-selection of the precursor ions in the classical MS2 mode.
- When considering the impact of the bait's presence on the structural characterization of target proteins, three scenarios are pertinent: intact mass and stoichiometry analysis of proteins and complexes (Type 1), intact mass and top-down analysis of individual proteins (Type 2), and subunit and topology analysis of protein complexes (Type 3).

For Type 1 analysis, the defined mass of the bait implies that its attachment does not influence the mass or stoichiometry analysis of the target proteins or protein complexes, thereby obviating the need for bait release. This is illustrated by three cases presented in MS1 spectra in Figures 2G, 2H and 3B, where the bait is significantly larger than the target, slightly smaller than the target, and significantly smaller than the target, respectively.

In Type 2 analysis, the bait can be released from the complexes through collision-based dissociation, which preferentially ejects a non-covalent subunit and does not compromise the integrity of the target protein. This is demonstrated in the examples of characterizing GFP variants after release of Ab or sdAb bait (Figure 2G and 2H) and avidin monomers after release of the biotin bait (Figure 3D and 3E).

In Type 3 analysis, the use of a small-sized bait ensures the integrity of the target complex. Protein complexes have a tendency to eject smaller and peripheral subunits during collisional dissociation³. Given that the bait-target interaction is typically situated peripherally within the bait-target complex, employing a smaller bait can facilitate its ejection under collision conditions that do not disrupt the target complexes. This concept is illustrated in the new **Figure S19**, where Hb was used as the bait for the target complex Hp(2)Ab(1). During collisional dissociation, the bait-target complex Hp(2)Hb(4)Ab(1) sequentially released Hb subunits across a range of collision energies, all the while maintaining the integrity of the target complex.

Even under extreme conditions, which may potentially trigger the release of a target subunit prior to the ejection of the bait, the stoichiometry analysis of the target complex remains unaffected due to the defined mass of the bait, which allows for accurate stoichiometric analysis regardless of the sequence of subunit ejection.

As for the potential conformational changes during the bait release process, since conformation characterization is not within the scope of this work, it does not pose a potential problem to this method.

In light of these considerations, we believe that neither bait attachment nor bait release restricts the prospective applications of SNaP-MS. We have added the corresponding discussion in the Results and Discussion sections in the revised manuscript.

- We concur with the reviewer that every method, including SNaP-MS, has its limitations. While the concerns raised by the reviewer can be addressed or mitigated, we acknowledge that inherent limitations persist. We have elaborated on some of these limitations, including the requirement for polymer removal and bait selection in Results and Discussion sections of the revised manuscript.

An example for inconsistency of the data can be seen in, Figure 5, where panel B demonstrates the presence of an intact proteasome generated from two separate plasmids, confirmed by native page, size exclusion chromatography, and electron microscopy. However, this intact proteasome is not detected by native mass spectrometry (panel C). This discrepancy between the detection of intact proteasome by alternative methods and its absence in native MS necessitates clarification and a convincing explanation to reconcile this disparity. Moreover, the absence of MS/MS analysis on the rearranged species further raises concerns. Validation is required to ensure that the “rearranged” a-subunit complex, is not actually a half proteasome containing the b-subunit pro-peptide.

- We express our gratitude to the reviewer for highlighting this concern. The discrepancy noted by the reviewer stems from the varying number of steps in the characterization workflows by MS and other techniques respectively. As detailed in the Results section, the 20S expressed from two separate plasmids demonstrated a lower resistance to degradation, a spontaneous process that includes the rearrangement of 20S subunits. In comparison to characterizations using native gel, SEC, and EM, native MS characterization necessitates an additional buffer exchange step, during which the extent of degradation escalates. Despite the detection of a higher extent of degradation of 20S expressed using this specific strategy in native MS than in characterizations with other techniques, native MS not only reveals different degradation extents of 20S expressed using different strategies due to their varying resistances to

degradation, but also simultaneously identifies the degradation products in detail (Figure 5D and 5E).

Therefore, SNaP-MS not only offers a method for screening optimal conditions for target protein expression but also provides a straightforward means of unveiling the molecular events underlying the dynamic evolution of the target protein system. We have enhanced the discussion on this topic in the Results section of the revised manuscript.

- We also appreciate the reviewer's suggestion to validate the identification of degradation products with MS/MS data. In response, we have performed such analyses of the degradation products, including the rearrangement products, to confirm their identity (the new Figure S23 in SI). The fragmentations of the mass-selected degradation products unambiguously verified that these products were 7-mer and 14-mer of the α subunit, respectively.

Another issue of concern is the lack of novelty of the data assignment technique (Figure S14), which has been widely employed in the field for several years (PMID: 18314965). Consequently, this aspect fails to make a significant contribution to the advancement of the field.

- We express our gratitude to the reviewer for this comment, which prompted us to enhance the presentation of our descriptions and discussions to avoid any potential misunderstandings.

Indeed, the process of releasing a subunit from protein complexes through collision-based dissociation and the associated asymmetric charge partitioning, which leads to the charge reduction of residual subcomplexes, has been previously reported in several articles, including the one mentioned by the reviewer (PMID: 18314965). We did not claim novelty regarding this phenomenon. Instead, we incorporated components that exploit this phenomenon into our design of SNaP beads.

The novelty in the context of tandem MS is associated with the bait, which was initially modified on the bead surface and subsequently attached to the target protein complexes upon bead dissolving. This bait serves as a charge carrier and mass corrector for heterogeneous glycoprotein systems at the tandem MS stage.

Two challenges in characterizing heterogeneous glycoprotein systems are the resolution of protein signal charges and the accuracy of mass determination. The bait-target complexes, being the direct product of the bead dissolving step, can release the bait in tandem MS. The baits serve two purposes: (1) they remove charges from the target complexes upon their release, thereby improving the charge resolution of the residual subcomplex signals; and (2) they provide an additional constraint for mass determination based on mass balance, *i.e.*, the total mass of bait-target complexes equals the sum of the bait mass and the target mass, thereby improving the accuracy of mass determination. Given that the accuracies of both the bait-target mass and the target mass are compromised by the inconsistent mass proteins of the corresponding ions in different charge states, the bait, which is a homogeneous molecule with a defined mass, can serve as a corrector to improve the mass determination for the targets based on the mass balance rule (refer to the detailed description in the caption of the new Figure S18C).

The design of SNaP beads allows for the attachment of such baits, which are neither endogenous in the target protein system nor introduced in an extra step between bead

dissolving and MS characterization, to the targets. This design provides a convenient means to improve the accuracy of mass determination of heterogeneous glycoproteins, without the requirement for special models of mass spectrometers equipped with electron- or proton-based charge manipulation function or CDMS function.

Since neither such baits (serving both as a charge carrier and mass corrector) nor the method of applying them to the target system has been previously reported, we believe the novelty related to this function is valid.

- We have rewritten the corresponding paragraphs in both the Results and Discussion sections for improved clarity.

Several inaccuracies in the references cited throughout the manuscript were identified, which diminish the overall credibility of the work and require correction.

- We express our gratitude to the reviewer for highlighting this oversight. Upon careful examination of the citation of references, we found that the inaccuracies were due to the incompatibility of different reference management software used by various authors in the original version of the manuscript. We have rectified these errors in the revised version.

Additionally, the manuscript lacks complete data on the theoretical and measured masses, including associated errors, for all species depicted in the figures. This crucial information is necessary to evaluate the accuracy and reliability of the reported results.

- We appreciate the reviewer's suggestion to include these data. In response, we have incorporated an additional supplementary table (Table S2), in the revised SI.

Reviewer 4

The manuscript describes an exciting technology for protein purification in the native state. The work addresses a fundamental problem in protein purification/pulldown workflow, tags that are extremely high-affinity and provides high-fidelity in substrate capture are often irreversible. This leads to difficulty in eluting the substrate from the bait post-pulldown, especially under native conditions. The authors here claim to solve this problem by developing universally applicable chemically/photochemically dissolvable microbeads, termed SNaP. I have several major concerns/questions about the claims the authors have made. However, if these concerns are addressed satisfactorily in the revised version, I would strongly support the publication of this manuscript. Below are my specific concerns.

- We extend our deepest appreciation to the reviewer for their meticulous review and favorable and constructive feedback, which have significantly contributed to improving the quality of our manuscript.

General comment:

1. In some places, the authors veer off in discussing aspects that are not directly related to the question at hand. For example, in the second paragraph page 2, on glycosylation. Detecting well-resolved MS from heterogenous PTM samples is a major technical challenge in native MS. But that does not have anything to do with purification. An epitope that recognizes all proteoforms will carry the same problem, irrespective of how efficient the enrichment strategy is. A good example is the authors' case study with the Hp-Hb system. An efficient enrichment did not solve the mass heterogeneity problem. So, discussing this in the middle of the intro breaks the flow.

- We express our gratitude to the reviewer for their suggestion to improve the logical flow of our manuscript.
- Although the purification process itself does not resolve the issue of mass heterogeneity, both the targets of interest in biological samples, and the unique design of SNaP beads do.

As the most abundant and complex type of PTM, glycosylation contributes to a variety of biological processes and is prevalent in natural proteins and their complexes in biological samples. The difficulty in interpretation of the spectra of heterogeneously glycosylated proteins, which is due to the heterogeneity-induced peak broadening and signal overlapping, is another impediment to the direct structural characterization of natural proteins.

SNaP beads allow attachment of bait molecules, which have a defined mass, to the target proteins upon bead dissolving. The release of these baits at the tandem MS stage provides a convenient means to enhance the accuracy of mass determination for heterogeneous glycoproteins, without the need for specialized mass spectrometer models equipped with electron- or proton-based charge transfer function or CDMS function.

Characterizing heterogeneous glycoprotein systems presents two challenges: the resolution of protein signal charges and the accuracy of mass determination. The bait-target complexes, being the direct product of the bead dissolving step, can release the bait in tandem MS. The baits serve two functions: (1) they remove charges from the target complexes upon their release, thereby improving the charge resolution of the residual subcomplex signals; and (2)

they provide an additional constraint for mass determination based on mass balance, *i.e.*, the total mass of bait-target complexes equals the sum of the bait mass and the target mass, thereby enhancing the accuracy of mass determination. Given that the accuracies of both the bait-target mass and the target mass are compromised by the inconsistent mass proteins of the corresponding ions in different charge states, the bait, which is a homogeneous molecule with a defined mass, can serve as a corrector to enhance the mass determination for the targets based on the mass balance rule (the new Figure S18C).

These baits, initially conjugated on the bead surface and subsequently attached to the target protein complexes upon bead dissolving, are neither endogenous in the target protein system nor introduced in an extra step between bead dissolving and MS characterization. Therefore, the workflow of SNaP-MS inherently provides a solution to the heterogeneous mass problem through its bait-attachment and bait-release processes.

- In response to the reviewer's suggestion, we have revised the Introduction section to improve the logical flow, and added relevant discussion in Results and Discussion sections.

2. The overall strategy of synthesizing these beads seems very straightforward. If the authors' scientific claims are right, then this would certainly be a major step forward from the standard agarose beads used. So for the widespread use of the Authors' protocol, I recommend both the synthesis, as well as microbead preparation sections be further elaborated with detailed preparation protocols, troubleshoot, and checkpoints. That would also certainly broaden the applicability of the current work/strategy.

- We concur with the reviewer that the inclusion of such details is crucial. Consequently, we have revised the Methods section to incorporate comprehensive protocols and checkpoints for material synthesis and microbead preparation.

GFP-capturing experiment:

1. Here the authors used thiol-linked GFP-antibody for pulldown and subsequent reduction for elution. The manuscript does not describe how the antibody was modified. The SI section only details the DBCO modification, not the modification by the thiol-linked linker. Since the authors are using this as a benchmark to showcase the advantages of their approach, details of this, including prior references from which this strategy was adopted must be included.

- We concur with the reviewer that the inclusion of such details is crucial. We have incorporated the experimental details of thiol-linked protein modification in the revised Methods section.

2. It is also a bit surprising that the authors used GFP-antibody here as the gold standard to pull down GFP. In practice, this is not right. The most used bait to pull down GFP is the variable domain GFP-nanobody. Is there a reason why that was not used?

- We appreciate the reviewer's suggestion. We have conducted additional experiments using an anti-GFP single-domain antibody (sdAb) as the bait. We have carried out purification using a variety of beads modified with different baits, including sdAb, and quantitatively evaluated the performance of these baits. As demonstrated in revised Figure 2, the beads modified with sdAb showed a similar efficiency to those modified with the antibody.

Synthesis of dissolvable beads:

1. Overall the synthesis section is difficult to follow. A general comment is to write the name of the chemicals used in the steps so that non-specialist users can follow the chemistry. A good example of that is Figure S3. Please provide the name of the chemicals on top of the arrow. Also refer to these figures, while describing the synthesis process in the earlier SI-section (between pages S2-S7).

- We express our gratitude to the reviewer for their suggestion. In response, we have thoroughly revised the corresponding paragraphs in the Methods section, which has now been moved from the SI to the main text. The revision includes the specific names of the chemicals used in the synthesis procedure.
- We have reorganized the materials from the original Figure S3 into new Figures S6 and S7. In addition, we have included the names of the chemicals above the arrows and numbered the steps. This numbering facilitates citation to the corresponding steps in the protocol description.
- We have incorporated citations to the corresponding steps illustrated in the new Figures S6 and S7 in the revised paragraphs of the Methods section.

2. While adequate NMR spectra have been shown to confirm the synthesis of the bead material, there is little quality control shown on the microbeads themselves. At least some TEM images of the beads and comparison with the commercial beads would be necessary.

- We agree with the reviewer's suggestion regarding the characterization of the casted beads. After making effort to acquire the EM images of the SNaP beads, we recognized that due to the inherent properties of hydrogel materials, including high water content and swellability, the images obtained under vacuum conditions do not accurately represent their functional morphology in solution. Consequently, we transitioned to particle analysis of fluorescence microscopic images for the characterization and quality control of the fabricated beads. The new Figure S5 presents frequency distribution histograms of the diameters of both chemically dissolvable SNaP beads and photo-dissolvable SNaP beads.

3. Figure S3 legends need to be expanded. It is a key figure.

- We concur with the reviewer's feedback regarding the caption of the original Figure S3. The materials from this figure have been restructured into the new Figures S6 and S7. In response to the reviewer's suggestion, we have elaborated on the captions for these figures.

4. The fabrication step of the microbeads and any related quality control should be elaborated.

- We express our gratitude to the reviewer for their insightful comment. In response, we have elaborated the fabrication of SNaP beads and the associated quality control in the Methods section of the revised manuscript.

Strep-GFP-capturing experiment from TAMARA labeled lysate:

1. Line 151/152 should cite Figure S3 for the GFP-strep modification protocol

- We have cited the corresponding steps illustrated in the original Figure S3, which has been repositioned as Figure S6, at the appropriate locations within the text.

2. Again, the authors need to justify why they are using GFP-antibody, instead of the more commonly used GFP-nanobody; especially when protocol is available to couple nanobody to beads using thiol or other chemistry (PMID: 26633879)

- As mentioned in our response to a previous comment, we have conducted additional experiments using an anti-GFP single-domain antibody (sdAb) as the bait. After purifying GFP from E. coli lysate using beads modified with sdAb baits, we employed bottom-up proteomics to quantitatively evaluate the performance. This was achieved by measuring the ratio of peptides from the captured GFP to those from the background proteins, as shown in the new Figure 2.

3. Figure 2A follows what is expected based on the binding affinities. But that has nothing to do with the developed SNaP beads. The real power of the technique is that it can use the highest affinity binding to capture the target with the least background, which other agarose beads also can, but then can subsequently elute it in native conditions, which one cannot do from standard agarose functionalized beads. So the “money-shot” here is not the comparison between different tags with the same SNaP beads but between the highest affinity elutable tag in agarose beads vs. the highest affinity tag (otherwise non-elutable) in SNaP beads. Among the tags used here, streptavidin is the highest affinity tag that allows native elution using desbiotin from standard agarose beads. Streptavidin, while offering the highest affinity, cannot be eluted from agarose. So, the authors should compare the capture/background ratio between Streptavidin-SNaP and Streptactin-Agarose. Part of that data is in Figure S9. While that could stay there, Figure 2 should include a similar fl.-based GFP/background capture ratio for Streptactin-agarose.

- We express our gratitude to the reviewer for their suggestion and for emphasizing the key advantage of this method. We have conducted additional bottom-up proteomics experiments to quantitatively evaluate the performance. This was achieved by measuring the ratio of peptides from the captured GFP to those from the background proteins for both SNaP beads and agarose beads. Furthermore, we have updated Figure 2 to include these data, as well as the fluorescence data from the original Figure S9.

4. Following the same notion, the authors should compare side-chain cleavable solid agarose beads having the Streptavidin bait.

- We concur with the reviewer’s suggestion. In response, we have conducted additional bottom-up proteomics experiments to quantitatively evaluate the performance. This was achieved by measuring the ratio of peptides from the captured GFP to those from the background proteins for M280 streptavidin-modified beads, which are side-chain cleavable. Furthermore, we have updated Figure 2 to include these data.

5. Both here, as well as in subsequent sections, the authors have used native MS to demonstrate the fidelity of the native elution process (by showing non-covalent homomeric and heteromeric complexes), as well as the purity of the eluted. While native MS can certainly confirm the first, it is not the most suitable approach for demonstrating the second. This is because in native MS it is feasible to tune MS in a manner that makes it more sensitive towards a particular m/z region. In UHMR, the

instrument used here, change in trapping gas Pressure and/or rf voltages can significantly bias the spectra towards high/low mass region. So for assessing the quality of the purified samples, conventional bottom-up proteomics to see how target proteins are enriched over the background can still provide the most in-depth view of the purification process. So to fully convince readers to switch from their established agarose bead approach to SNaP beads a simple LFQ-based bottom-up proteomics approach will go a long way. The authors can take Streptavidin-SNaP eluted samples and Streptactin-Agarose eluted samples. Using an LFQ-based study can show the degree of enrichment of GFP over the background in both cases. This will also broaden the use of the method, as there are certainly more labs doing pull-down proteomics to find interactors than doing nativeMS. A very high-affinity elutable tag would certainly benefit that endeavor too.

- We express our gratitude to the reviewer for their comments. As mentioned in our responses to the previous two comments, we have conducted label-free quantitative proteomics for various types of beads, including SNaP beads, as shown in the updated **Figure 2**. Additionally, we have performed LFQ analysis to assess the performance of SNaP beads modified with different baits, as depicted in the new **Figure 2**.

Compatibility of recovered protein sample with native MS characterization

1. This is an important point, as it seems eventually for native MS one has to do another round of purification. Is the SEC a must? Zeba/bio spin-based spin columns are typically used for quick buffer exchange from Tris/Phosphate buffers to native MS-compatible AmAc. Can they not remove the polymer? One exciting prospect of this high-affinity/high-efficiency capture of SNaP beads is that one can purify native MS-worthy material from a very small amount of starting samples. However, having SEC as a necessary step in this pipeline makes it slightly more sample-demanding. It would be nice if authors could circumvent this need.

- We concur with the reviewer's suggestion for a clearer presentation of polymer removal.

Indeed, the removal of polymers is a necessary step for native MS characterization of the purified products. Depending on the properties of the samples and the objectives of the characterization, several approaches can be employed for this task. In addition to SEC, these approaches include (1) membrane-based centrifugal buffer exchange, (2) ion-exchange chromatography (IEX), and (3) collisional fragmentation at the MS stage. In this study, MS data for avidin samples were acquired through Approach (1), those for 20S and haptoglobin samples were acquired through Approach (2), and those for GFP samples were acquired through Approach (3) following SEC.

We attempted to use a bio-spin column for polymer removal. However, the results were not satisfactory, which we attribute to the inadequate retention of polyacrylamide by the material of the bio-spin column.

We have updated the description and discussion on this topic in the revised Results section for improved clarity.

2. Also, SEC itself is a protein purification process. So if it is a must step, then the comparison shown in Figure 2 has to be done at the end SEC of the respective samples.

- As explained in our response to a previous comment, SEC is not a mandatory step. It can be replaced with membrane-based centrifugal buffer exchange, IEX, or collisional fragmentation at the MS stage. The purpose of presenting data in Figure 2 includes comparing the performance of different types of baits modified on the bead surface. Including an additional SEC purification process might introduce further complications. Moreover, incorporating an SEC step could potentially hinder comparison with results from purifying other protein systems that employ different polymer removal approaches. Therefore, after careful consideration, we have decided not to include SEC in the workflow for evaluating the purification performance.

Avidin-capturing experiment

1. For r-Avidin (figure 3B), the monomer seems to be emerging at a higher in-source. Is that because background polymer requires more in source for better desolvation?

- In the experiment presented in Figure 3B, a higher in-source fragmentation energy (SF) is required for the intentional release of the monomer, rather than for improving desolvation. In this context, we acquired a set of native MS spectra of avidin samples using different SF settings. Lower SF settings were used to maintain the integrity of protein assemblies for the characterization of intact complexes, while higher SF settings were intentionally used to release the monomers for mass measurement and subsequent top-down analysis. As illustrated in the bottom spectra acquired with SF 5, adequate desolvation of the intact complexes can be achieved under gentle conditions that do not result in monomer release. The observed mass increase of the recovered avidin, compared to the original avidin, was due to the incorporation of baits modified with the linker moiety.
- We have updated the corresponding description in the Results section to clarify this point and avoid potential confusion.

2. Overall, the peaks look rather broad, especially for tetramer. The authors said that they used in-source energy. UHMR also has a front-end trapping-based activation, termed desolvation energy, which is more effective than standard insource. Have authors tried using it to see if they could get better-resolved spectra?

- We express our gratitude to the reviewer for this highly professional suggestion. Indeed, we have optimized several parameters, including desolvation energy, to ensure the quality of the spectra. The broad peaks of the avidin tetramers shown in Figure 3B are the result of mass heterogeneity, not suboptimal desolvation.
- As illustrated in the proteoform-resolved spectra shown in Figures 3D and 3E, an *r* avidin monomer exhibits a high degree of microheterogeneity and a broad mass distribution. This mass distribution range is multiplied when these monomers assemble into tetramers. The detected mass distribution widths of the intact tetramers and trimers, which were products of monomer release at higher SF, align with our simulation results based on the mass distribution width of the monomers. This suggests that their signal peak widths did not contain contributions from inadequate desolvation. Further evidence supporting this conclusion is that even under harsh conditions where higher SF resulted in dissociation of the

tetramers, their peak widths remained unchanged. We have updated the corresponding description in the Results section to clarify this point.

3. Due to possible effects on the intramolecular disulfide bonds, I thought this was the perfect example to use the UV-dissolvable beads. The authors mentioned making it in the first section but never used it. If they want to claim that they have also made UV-dissolvable SNaP beads, they need to use them here.

- We express our gratitude to the reviewer for this suggestion. In response, we have conducted a capture test as a proof of concept using UV-dissolvable beads. The results of this test are presented in the new Figure S9B.

Hp-Hb capturing experiment

1. In Figure 4B, the charge state assignment should be made to the MS. In P3, the peaks are not resolved enough to annotate the charge state assignment.

- We concur with the reviewer's suggestion regarding the labeling of peaks with charge state assignments in Figure 4B. We have revised this figure to include such annotations.
- Hp purified from P3 exhibited a higher degree of oligomerization than samples from other individuals. This increase in size and accumulation of heterogeneity compromised the resolution of signal clusters corresponding to larger oligomers in different charge states, resulting in intrinsically highly convoluted signals in the MS1 spectrum. This is independent of the resolving power of the mass spectrometers, experimental methods, or data acquisition parameters. However, we were able to measure the masses of such large complexes using a tandem-MS based approach, as detailed in Figure S18C and its caption.

2. Here, the complexity, arises from glycosylation. Since the SNaP tag allows native-complex binding, would it not be possible to do a de-glycosylation, on the bead by enzyme such as PNGase? Post-reaction, the beads can be separated from the free glycan and PNGase by centrifuging. Subsequently dissolved and subjected to MS.

- We express our gratitude to the reviewer for this suggestion. Indeed, while PNGase is an effective enzyme for removing N-glycans, its deglycosylation efficiency is highly dependent on substrate accessibility. For proteins with multiple N-glycosylation sites that are not sufficiently exposed, PNGase functions less effectively under native conditions and typically requires denaturing conditions to ensure efficient deglycosylation. We have conducted a deglycosylation experiment where we used PNGase F to treat the Hp 2-2 sample under native conditions. As depicted in the new Figure S17, such treatment resulted only in modest deglycosylation and did not significantly enhance signal resolution.
- The reduced effectiveness of glycosidase in reducing heterogeneity caused by complex glycosylation under native conditions highlights the value of the tandem MS-based approach in improving charge resolution and mass accuracy through bait attachment. This is a unique function provided by the SNaP-MS workflow.

References

1. Langlois MR, Delanghe JR. Biological and clinical significance of haptoglobin polymorphism in humans. *Clin Chem* **42**, 1589-1600 (1996).
2. Braeckman L, De Bacquer D, Delanghe J, Claeys L, De Backer G. Associations between haptoglobin polymorphism, lipids, lipoproteins and inflammatory variables. *Atherosclerosis* **143**, 383-388 (1999).
3. Delanghe J, Langlois M, Duprez D, De Buyzere M, Clement D. Haptoglobin polymorphism and peripheral arterial occlusive disease. *Atherosclerosis* **145**, 287-292 (1999).
4. Delanghe JR, *et al.* Haptoglobin Polymorphism and Complications in Established Essential Arterial-Hypertension. *Journal of Hypertension* **11**, 861-867 (1993).
5. Naryzny SN, Legina OK. Haptoglobin as a Biomarker. *Biochem Mosc Suppl B Biomed Chem* **15**, 184-198 (2021).
6. Loo JA, *et al.* Electrospray ionization mass spectrometry and ion mobility analysis of the 20S proteasome complex. *Journal of the American Society for Mass Spectrometry* **16**, 998-1008 (2005).
7. Ma X, Loo JA, Wysocki VH. Surface induced dissociation yields substructure of *Methanosarcina thermophila* 20S proteasome complexes. *International Journal of Mass Spectrometry* **377**, 201-204 (2015).
8. Sharon M, Witt S, Glasmacher E, Baumeister W, Robinson CV. Mass Spectrometry Reveals the Missing Links in the Assembly Pathway of the Bacterial 20 S Proteasome. *Journal of Biological Chemistry* **282**, 18448-18457 (2007).

REVIEWER COMMENTS

Reviewer #1 (Remarks to the Author):

The author addressed all the main concerns. It's very close to publication before addressing several minor points below.

Figure S24B, the y-axis should be marked as "artificial units".

Figure S25A, the scale bar should be specified, do not leave an unlabelled scale bar.

Reviewer #2 (Remarks to the Author):

I appreciate the significant work performed and the changes made by the authors during the revision, which have addressed most of my previous detailed technical comments. However, some of main concerns about the claims of the papers have not really been addressed and some issues with presentation could also be improved.

1)The claim on sensitivity: The authors mostly tried to argue about how “sensitive” is what has been shown currently. But 100 microliter is a volume routinely analyzed in the MS community now. The quoted 10 ng/microliter is also a rather high concentration for detecting proteins (at least 4 orders of magnitude less sensitive than ELISA). These are not sensitive at all -- perhaps the authors are sometimes confusing sensitivity with specificity, which has been shown here because of the use of antibody bait. Specifically for the example of the highly abundant 20S proteasome, the community has extensively studied endogenous 20S proteasome using native MS over the last 2 decades without having to resort to the technique here (for example, see a review: <https://doi.org/10.1016/bs.mie.2018.12.029>). A native characterization of a much less abundant endogenous protein complex was reported recently (Nature Communications 14, 8400 (2023)), which is more sensitive than what is demonstrated here but sensitivity was not claimed there (and rightfully so).

Therefore, please remove the claim of “sensitive” from the manuscript title and the text.

2) Even though the nice advances made by the authors are well appreciated, for similar reasons as discussed above, the general claim on “complex biological samples” and clinical samples is not well supported by the current results, Please remove or significant tone down such claims. For example, the last sentence “serve as a general strategy for characterizing natural protein complexes in biological and potentially clinical studies”.

3) Also, this newly added sentence around line 435 should be further revised due to the recent native proteomics paper above. "The approaches to isolate or enrich low-abundant proteins in biological samples using nanomaterials^{49, 50, 51} are not compatible with native conditions."

4) The Figure 1 scheme remains messy and ineffective. The authors have consolidated all details into a single figure, making it difficult for readers to discern the key design concepts and innovations. The authors have gone too far in the direction of adding details. It is suggested to separate the details of material synthesis from the material design and functionality (i.e. how are the materials used for protein separation, elution, and proteomic analysis etc.). Additionally, I suggest the experimental processes to be presented in a little bit more details for the protein separation experiments involving the three proteins in subsequent sections.

5) The author describes two designs of beads herein, one utilizing disulfide bonds that can be cleaved using certain reducing agents, and the other employing photo-dissolvable chemical bonds that can be cleaved using ultraviolet light. The latter design is intended to prevent damage to the native structure of proteins caused by reducing agents. However, in the experiments after Figure 3, the authors do not emphasize which design was used for native protein enrichment. Please add explanations (or provide a comparison between the functionalities of the two types of beads for native proteomics). The authors also claim that 5 mM TCEP can cleave disulfide bonds, suggesting that this method should have minimal impact on the native structure of proteins. If the authors are trying to say both approaches could help native proteomics, I suggest that the author show a comparison of the native mass spectrometry data for proteins eluted using both methods.

Reviewer #4 (Remarks to the Author):

The authors have made a lot of effort in revising their manuscript based on the suggestions provided. I have two minor comments:

1. Authors keep referring to in-source fragmentation as SF. UHMR has two different ways of providing in-source fragmentation. One is traditional front-end activation at the S-lens and the next is between the flatapoles. The latter is termed as Desolvation energy. Authors in their response mentioned that they used desolvation but in the manuscript wrote "in-source". It would be useful to clarify that. This is because, during the application of Desolvation energy, ions are trapped for 4ms. This is a unique feature of UHMR. So it would be necessary to know/clarify whether or not that is necessary. Most nMS only have traditional in-source fragmentation that does involve trapping.

2. Next, while the authors did provide a gel from the UV-cleaved polymer, no MS is shown for the polymer. So if possible, it would be useful to provide proof that the UV cleavable polymer is also nMS compatible. But I would not hold this against delaying the publication

This is overall an impressive piece of work that is expected to progress the field of affinity capture forwards

Kallol Gupta

DETAILED POINT BY POINT RESPONSE

Reviewer 1

The author addressed all the main concerns. It's very close to publication before addressing several minor points below.

- We appreciate the reviewer's positive feedback. We have addressed the minor points as detailed below.

Figure S24B, the y-axis should be marked as "artificial units".

- We have incorporated a title for the y-axis and defined the unit as "artificial unit".

Figure S25A, the scale bar should be specified, do not leave an unlabeled scale bar.

- We have denoted this scale bar as 50 nm.

Reviewer 2

I appreciate the significant work performed and the changes made by the authors during the revision, which have addressed most of my previous detailed technical comments. However, some of main concerns about the claims of the papers have not really been addressed and some issues with presentation could also be improved.

- We express our gratitude to the reviewer for the constructive comments. We have revised the relevant statements and addressed these issues in accordance with the reviewer's comments. Please refer to our responses to the corresponding comments for more details.

1)The claim on sensitivity: The authors mostly tried to argue about how "sensitive" is what has been shown currently. But 100 microliter is a volume routinely analyzed in the MS community now. The quoted 10 ng/microliter is also a rather high concentration for detecting proteins (at least 4 orders of magnitude less sensitive than ELISA). These are not sensitive at all -- perhaps the authors are sometimes confusing sensitivity with specificity, which has been shown here because of the use of antibody bait. Specifically for the example of the highly abundant 20S proteasome, the community has extensively studied endogenous 20S proteasome using native MS over the last 2 decades without having to resort to the technique here (for example, see a review: <https://doi.org/10.1016/bs.mie.2018.12.029>). A native characterization of a much less abundant endogenous protein complex was reported recently (Nature Communications 14, 8400 (2023)), which is more sensitive than what is demonstrated here but sensitivity was not claimed there (and rightfully so).

Therefore, please remove the claim of "sensitive" from the manuscript title and the text.

- We acknowledge that the claim regarding sensitivity could potentially lead to misunderstandings. In response to the reviewer's suggestion and to avoid confusion, we have

removed such claim and modified the full name of the developed workflow, which previously included the term “sensitive”.

2) *Even though the nice advances made by the authors are well appreciated, for similar reasons as discussed above, the general claim on “complex biological samples” and clinical samples is not well supported by the current results, Please remove or significant tone down such claims. For example, the last sentence “serve as a general strategy for characterizing natural protein complexes in biological and potentially clinical studies”.*

- Following the reviewer’s suggestion, we have moderated the claims regarding the complexity or the clinical potential. We have eliminated all instances of “complex” and “clinical” from the attributes of “samples” in this manuscript.

3) *Also, this newly added sentence around line 435 should be further revised due to the recent native proteomics paper above. “The approaches to isolate or enrich low-abundant proteins in biological samples using nanomaterials^{49, 50, 51} are not compatible with native conditions.”*

- We are grateful to the reviewer for helping us correct this statement by citing this reference, which was published subsequent to the initial submission of our manuscript. We have revised the corresponding sentence as follows: *“The approaches to isolate or enrich low-abundant proteins in biological samples using nanomaterials either are incompatible with native conditions {cite Reference 49, 50, 51} or requires elution buffer specifically designed for preservation of non-covalent interactions {cite Nature Communications 14, 8400 (2023)}”.*

4) *The Figure 1 scheme remains messy and ineffective. The authors have consolidated all details into a single figure, making it difficult for readers to discern the key design concepts and innovations. The authors have gone too far in the direction of adding details. It is suggested to separate the details of material synthesis from the material design and functionality (i.e. how are the materials used for protein separation, elution, and proteomic analysis etc.). Additionally, I suggest the experimental processes to be presented in a little bit more details for the protein separation experiments involving the three proteins in subsequent sections.*

- We agree with the reviewer’s opinion on how to effectively illustrate the workflow. We have revised **Figure 1** by relocating all molecular structures to **Figure S1, S6 and S7**, and labeled the key steps with concise descriptive phrases to improve the readability.
- We have introduced the new **Table S1** in the SI to specify separation methods for proteins, and added more details on protein separation under the subheading of *Liquid chromatography (LC)* in the Experimental Section as suggested by the reviewer.

5) *The author describes two designs of beads herein, one utilizing disulfide bonds that can be cleaved using certain reducing agents, and the other employing photo-dissolvable chemical bonds that can be cleaved using ultraviolet light. The latter design is intended to prevent damage to the native structure of proteins caused by reducing agents. However, in the experiments after Figure 3, the authors do not emphasize which design was used for native protein enrichment. Please add explanations (or provide a comparison between the functionalities of the two types of beads for native proteomics). The authors also claim that 5 mM TCEP can cleave disulfide bonds, suggesting that this method should have minimal impact on the native structure of proteins. If the authors are trying to say both*

approaches could help native proteomics, I suggest that the author show a comparison of the native mass spectrometry data for proteins eluted using both methods.

- We thank the reviewer for helping us improve the description of experimental details. We have added a description in the revised Experimental Section to specify that these proteins were purified with chemically dissolvable SNAP beads. To demonstrate the negligible impact of the disulfide reduction condition for bead dissolving on the integrity of disulfide-containing protein complexes, we have compared the original dimeric Hp and SNAP-purified dimeric Hp, a protein containing multiple disulfide bonds. Both the intact mass and tandem mass spectra suggest that the integrity of not only the Hp dimer but also its complex with Hb were not affected by bead dissolving. We included this set of data in the new Figure S18A.

Reviewer 4

The authors have made a lot of effort in revising their manuscript based on the suggestions provided. I have two minor comments:

1. Authors keep referring to in-source fragmentation as SF. UHMR has two different ways of providing in-source fragmentation. One is traditional front-end activation at the S-lens and the next is between the flatapoles. The latter is termed as Desolvation energy. Authors in their response mentioned that they used desolvation but in the manuscript wrote "in-source". It would be useful to clarify that. This is because, during the application of Desolvation energy, ions are trapped for 4ms. This is a unique feature of UHMR. So it would be necessary to know/clarify whether or not that is necessary. Most nMS only have traditional in-source fragmentation that does involve trapping.

- Indeed, we have examined the effect of desolvation voltages (DV) on the detected tetramer signals. To avoid confusion, we have added a new figure (Figure S17) to demonstrate the mass increase of the detected recovered avidin tetramer is due to incorporation of bait rather than inadequate desolvation. The mass spectra in this figure were acquired with various DV settings. Pertaining to Figure 3B and the corresponding result description in the main text, we have replaced "SF" with "collisional dissociation" to emphasize our capability to switch the characterization target between tetramer and monomer through adjustment of collisional dissociation. Both the DV values and the in-source fragmentation values used in these experiments are specified in the figures or their captions.

2. Next, while the authors did provide a gel from the UV-cleaved polymer, no MS is shown for the polymer. So if possible, it would be useful to provide proof that the UV cleavable polymer is also nMS compatible. But I would not hold this against delaying the publication

- We appreciate the reviewer's comments on this issue. We have added the new Figure S13B to display the MS signals of UV-cleaved polymers, and the new Figure S14B to depict the removal of such polymers and the compatibility of such polymer removal with native MS. We have also investigated the influence of the structure of linear polymer or linker on the efficiency of bead dissolving and polymer removal for MS analysis, and have improved the workflow involving the photo-dissolvable SNAP beads. Given the current length of the manuscript and considering that the performance of photo-dissolvable SNAP beads can be

further optimized, and a detailed description of the chemical design and optimization may cause distraction, we decided to exclude the content on further optimization of photo dissolving from the scope of this manuscript.

This is overall an impressive piece of work that is expected to progress the field of affinity capture forwards.

- We express our gratitude to the reviewer for the favorable comments, which have significantly enhanced the quality of our manuscript.